# In-degree centrality in a social network is linked to coordinated neural activity

Elisa C. Baek [1✉], Ryan Hyon[1], Karina López [1], Emily S. Finn [2], Mason A. Porter [3,4] &
Carolyn Parkinson [1,5✉]

Convergent processing of the world may be a factor that contributes to social connectedness. We use neuroimaging and network analysis to investigate the association between the social-network position (as measured by in-degree centrality) of first-year university students and their neural similarity while watching naturalistic audio-visual stimuli (specifically, videos). There were 119 students in the social-network study; 63 of them participated in the neuroimaging study. We show that more central individuals had similar neural responses to their peers and to each other in brain regions that are associated with high-level interpretations and social cognition (e.g., in the default mode network), whereas less-central individuals exhibited more variable responses. Self-reported enjoyment of and interest in stimuli followed a similar pattern, but accounting for these data did not change our main results. These findings show that neural processing of external stimuli is similar in highly-central individuals but is idiosyncratic in less-central individuals.

[1] Department of Psychology, University of California, Los Angeles, Los Angeles, CA, USA. [2] Department of Psychological and Brain Sciences, Dartmouth College, Hanover, NH, USA. [3] Department of Mathematics, University of California, Los Angeles, Los Angeles, CA, USA. [4] Sante Fe Institute, Santa Fe, NM, USA. [5] Brain Research Institute, University of California, Los Angeles, Los Angeles, CA, USA. ✉email: elisabaek@ucla.edu; cparkinson@ucla.edu

Humans are incredibly social, and difficulties with social connection have been linked to myriad negative consequences, including increased likelihood of morbidity and mortality[1–4]. Having many social ties is one factor that can protect against the detrimental consequences of social isolation and disconnection[5–9]. Differences in the extent of social connectedness occur in many human social networks[10–12], and it has been established that such differences are critical determinants for the well-being of individuals[5]. They can also have far-reaching consequences for the social networks in which individuals are embedded. For example, central individuals often have significant influence in shaping the opinions and attitudes of social groups[13–16].

Despite robust evidence for the benefits to one's health and well-being of being well-connected and the fact that well-connected individuals are well-positioned to exert influence on others in their social networks, there are significant gaps in our understanding of which factors distinguish well-connected individuals (such as those with many friends) from other individuals. For instance, although some personality traits (such as extraversion and emotional stability) have been associated with being well-connected in some social networks[17,18], such links have not been found in other contexts[19–21]. It is possible that approaches that focus on personality do not capture features that distinguish central individuals across various social contexts. For example, one possibility is that individuals who occupy central positions in a social network process the world around them in a way that allows them to relate to, understand, and connect with a larger number of people in their communities. Recognizing and adhering to social norms is critical to being successful in forming and maintaining social ties[22], so well-connected individuals may be more attuned to their peers' norms either as a cause or as a consequence (or as a combination of both) of their central position in a network. Accordingly, well-connected individuals may process the world around them in ways that are very similar to their peers. Correspondingly, it is possible that individuals with fewer social connections (specifically, those with lower degree centrality, once one defines a social connection of interest) may process the world around them in ways that are less similar to their peers (including one another) than is the case for individuals with many social connections (i.e., those with higher degree centrality).

In the present paper, we test the hypothesis that individuals who occupy central positions in their social networks have neural responses to naturalistic stimuli (specifically, videos) that are more similar to those of their peers than individuals who occupy less-central positions. Specifically, we test whether individuals who many others nominate as a frequent social partner (i.e., who have a high in-degree centrality) have neural responses that are, on average, more similar to their peers than individuals who few people indicate as a frequent social partner and thus have a low in-degree centrality. There are many ways of defining the importance (i.e., centrality) of a node in a network[23,24]; as a shorthand, we use the term "highly central" to refer to having a high in-degree centrality. Measuring neural activity during a naturalistic paradigm (in which people view complex audiovisual stimuli, such as videos, that unfold over time) allows one to obtain insight into individuals' unconstrained thought processes as they unfold[25]. Coordinated brain activity between individuals (i.e., large inter-subject correlations (ISCs) of neural responses) during the viewing of dynamic, naturalistic stimuli has been associated both with friendship[26] (where, as in previous work on friendship networks[18], the definition of "friendship" was based on who nominated whom as a frequent social partner) and with shared interpretations and understanding of events[27–29]. Therefore, the extent to which an individual, on average, has similar neural-response time series as their peers can provide insight into the extent to which they process the world around them in a way that reflects the shared values, beliefs, and experiences of their communities.

We also test whether individuals who are highly central in their social networks are very similar to other highly-central individuals in how they process external stimuli, whereas less-central individuals process external stimuli in their own idiosyncratic ways. To help explain this idea, we draw an analogy from the famous opening line of the novel *Anna Karenina*[30]: "Happy families are all alike; every unhappy family is unhappy in its own way." An Anna Karenina principle posits that endeavors with particular outcomes share similar characteristics (so, in that sense, they "are all alike") and that a lack of any one of the characteristics results in the absence of the outcome in question[31]. The concept of an Anna Karenina principle has been applied to study various phenomena[32]. For example, it was used recently to link neural similarity with behavioral outcomes, such as trait paranoia[33]. In the present work, we test the hypothesis that "Highly-central individuals are all alike, but each less-central individual is dissimilar in their own way."

We first test whether highly-central individuals in a community process external stimuli in a way that is more similar to other community members than is the case for less-central individuals. We assess this idea by calculating the mean neural similarity between individuals and their peers. (See our participant-level ISC analysis in the Methods section for more details.) We also test the hypotheses that highly-central individuals have very similar neural responses to one another and that each less-central individual responds in their own unique way (i.e., idiosyncratically). Our results provide support for both hypotheses. We found that, on average, highly-central individuals had very similar neural responses to other members of their communities and especially to other highly-central individuals in brain regions that are associated with shared high-level interpretations and social cognition (e.g., regions of the default mode network). By contrast, we found that less-central individuals had comparatively idiosyncratic neural responses. We obtained similar results when we controlled for demographic similarities and social distances between individuals. Additionally, although participants' self-reported enjoyment of and interest in the stimuli followed a similar pattern as in the brain data, accounting for their self-reported preferences did not change our main results. Taken together, our findings suggest that individuals who are central in a social network tend to be more similar to one another in the ways that they process external stimuli than individuals who are less central and that each less-central individual is dissimilar in their own idiosyncratic way.

## Results

**Social-network characterization.** We characterized the social networks of individuals who live in two different residential communities of first-year students at a large state university (University of California, Los Angeles) in the United States. A total of 119 participants completed an online survey in which they indicated individuals with whom they socialized most frequently within their community (consistent with prior work on friendship networks[18]). (See the Methods section for further details.) Some of these participants also completed the functional magnetic resonance imaging (fMRI) part of the study. (The fMRI part of the study included $N = 63$ people after exclusions; see the Methods section.) Using the responses of the participants, we constructed a directed network for each of the two communities (see Fig. 1). In each of these networks, a node represents an individual and a directed edge represents one individual

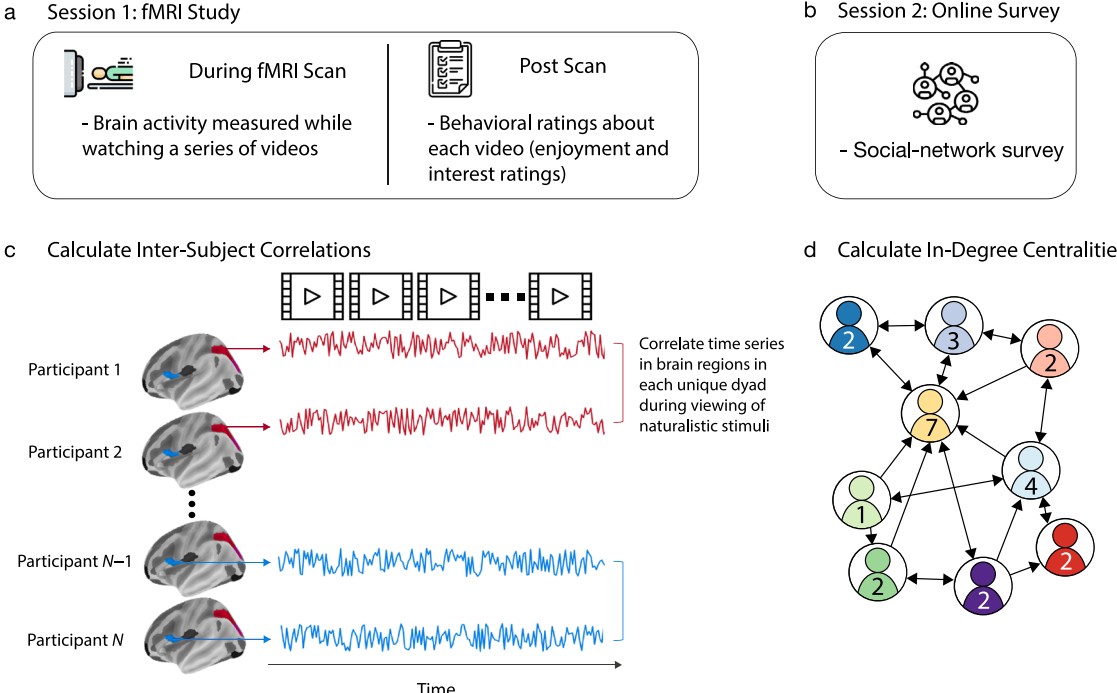

**Fig. 1 Study paradigm and calculations. a** Schematic of the fMRI study paradigm. In session 1 of the study, participants attended an in-lab session in which their brain activity was measured using fMRI while they watched a series of naturalistic stimuli (i.e., videos). After the fMRI scan, the participants provided ratings on how enjoyable and interesting they found each video. **b** Schematic of our social-network survey. In session 2 of the study, participants completed an online social-network survey in which they indicated the individuals in their residential community with whom they were friends. **c** Schematic of neural similarity. We extracted the time series of neural responses that were obtained as participants viewed the stimuli. We then calculated inter-subject correlations (ISCs) of these time series for each of 214 brain regions. **d** Schematic of our network calculations. Based on the participants' responses in (**b**), we constructed two directed, unweighted networks—with one for each residential community—in which each node represents an individual and each directed edge represents one individual nominating another as a friend. For each individual, we calculated in-degree centrality, which counts the number of times that that individual was nominated as a friend by others in their own residential community.

nominating another as a frequent social partner. For each individual, we calculated in-degree centrality, which counts the number of times that the individual was nominated as a regular social partner by someone else in the network. We chose to quantify an individual's centrality within their community in terms of in-degree centrality because it captures the extent to which others in the community consider the individual to be a regular social partner. Another advantage of in-degree centrality is that an individual's in-degree centrality (unlike some other measures of centrality, such as out-degree centrality) does not rely at all on one's own self-reported answers about the relationships that one has with others. Therefore, in-degree centrality is not susceptible to erroneous perceptions of one's own social partners and is less susceptible than other notions of centrality to mis-characterization of social ties due, for example, to any given participant's inattention during a survey or atypical interpretations of survey questions (because an individual's in-degree centrality is based on data that is aggregated across the responses of many participants). In-degree centrality is particularly suitable for our study because it is not affected by the presence of multiple components in a network, unlike most other measures of centrality (e.g., eigenvector centrality)[23].

In our primary analyses, we used a median split to binarize our sample into high-centrality and low-centrality groups. This choice is consistent with recent studies that related neural similarity with behavioral measures[34,35]. In our fMRI study, we classified participants as part of the high-centrality group if they had an in-degree that was larger than the median (specifically, if it was more than 2; there were $n_{high} = 23$ such people) and into the low-centrality group if they had an in-degree that was less than or

equal to the median (specifically, if it was less than or equal to 2; there were $n_{low} = 40$ such people). See Supplementary Fig. 1 for plots of the in-degree distributions. Because the median-split approach resulted in unevenly sized groups, we also conducted additional analyses to examine the relationships between the original non-binarized version of centrality and neural similarity whenever possible, as we describe in more detail below. We also conducted analogous exploratory analyses with approximately equal-sized groups by contrasting individuals with in-degree centralities in the top and bottom thirds of the distribution. This yielded similar results to our main findings. See Participant-level ISC analysis and Dyad-level ISC analysis for more details.

**Neural similarity**. During our fMRI study, participants watched 14 video clips that span a variety of topics (see Supplementary Table 1). We calculated ISCs of time series of neural responses that were measured with fMRI to capture shared neural responses across participants during the processing of naturalistic stimuli[36] (see Fig. 1). First, we extracted the mean-response time series across the entire video-viewing task from both (1) each of the 200 cortical regions in the 200-parcel version of the Schaefer et al.[37] parcellation scheme and (2) 14 subcortical regions[38]. (See the Methods section for more details.) This resulted in a total of 214 brain regions across the whole brain. For each of the 1952 unique pairs of participants (i.e., dyads) in our fMRI sample, we then computed the Pearson correlation between the dyad members' time series of neural responses for each cortical region. This yields one correlation coefficient per unique dyad for each brain region. See the Methods section for more details.

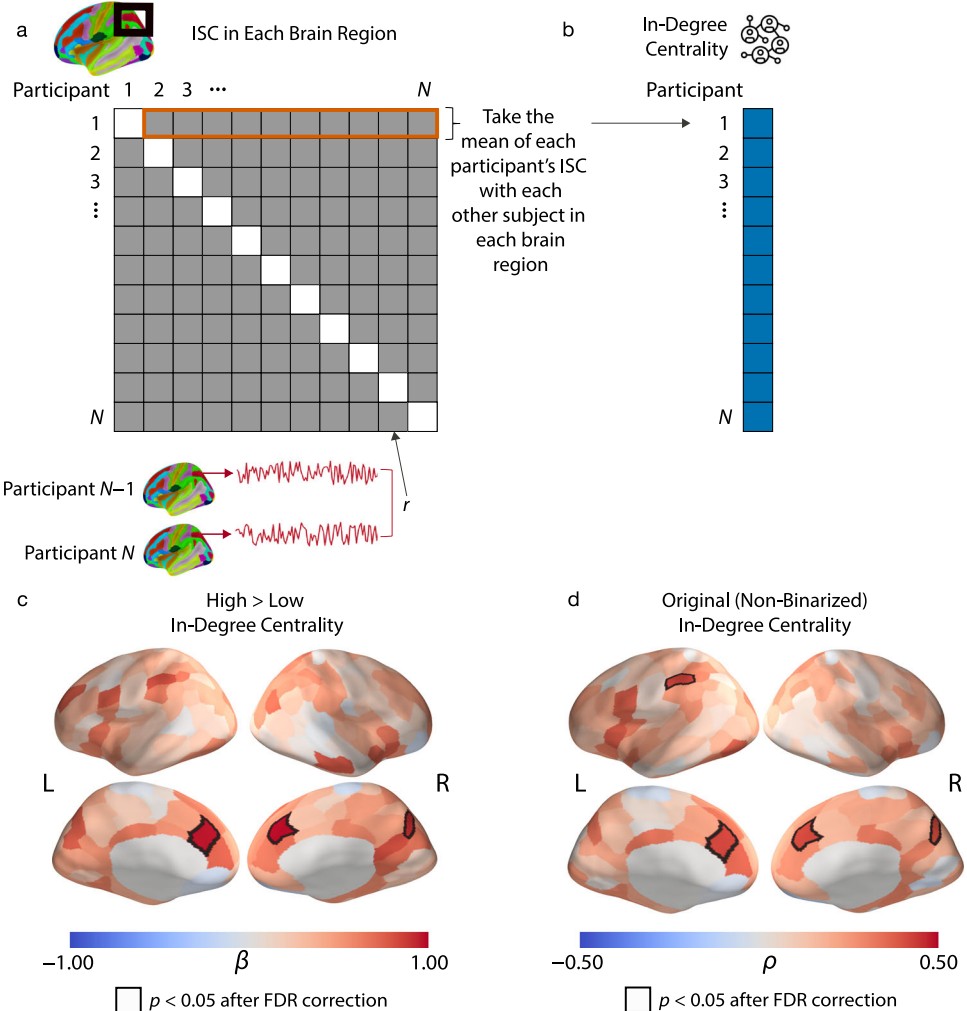

**Fig. 2 Participant-level analysis. a** Our approach for participant-level analysis. First, we Fisher $z$-transformed the dyad-level ISCs, which are encoded by a matrix of pairwise Pearson correlation coefficients (which we denote by $r$). We then computed the mean of each participant's ISC with each other participant. (In other words, we took the mean of each row of the matrix.) We performed the above calculations for each of the 214 brain regions. This yields one ISC value for each participant for each brain region. The ISC value encodes the mean similarity of the neural responses between the participant and each other participant in the corresponding brain region. **b** We tested for relationships between the participants' in-degree centralities and these participant-level ISC values in each brain region. **c** Our results that relate mean ISCs with the binarized in-degree centrality variable indicated that individuals with high in-degree centralities had much larger mean neural similarities with their peers in the bilateral DMPFC and precuneus than individuals with low in-degree centralities. **d** Our results that relate mean ISCs with the original (i.e., non-binarized) in-degree centrality values gave similar results as the analysis in (**c**). We found that the mean ISCs in the bilateral DMPFC, precuneus, and the superior parietal lobule were positively correlated with in-degree centrality. The quantity $\beta$ denotes the standardized regression coefficient, and $\rho$ denotes the Spearman rank correlation. All results are FDR-corrected at $p < 0.05$, which corresponds to an uncorrected $p$-value of 0.009 in (**c**) and an uncorrected $p$-value of 0.001 in (**d**). All of the reported $p$-values are two-tailed. Source data are provided as a Source Data file.

**Participant-level ISC analysis.** We tested whether individuals who had higher in-degree centralities in their communities exhibited more typical neural responses than individuals with lower in-degree centralities. To do this, in each brain region, we transformed our dyad-level neural similarity measure to a participant-level measure by calculating the mean Fisher $z$-transformed[39] ISC value for each participant with each other participant. This yields one ISC value for each individual for each brain region; this value encodes a mean similarity of the neural responses between the individual and all other individuals in the corresponding brain region (see Fig. 2a). After calculating these values, we fit one generalized linear model (GLM) for each brain region with the ISC in the respective brain region as the dependent variable (which we transformed into $z$-scores to produce standardized coefficients) and the binarized in-degree centrality as the independent variable (see Fig. 2b). Finally, we employed

false-discovery rate (FDR) correction to correct for multiple comparisons across brain regions. We found that high in-degree centrality was associated with larger mean neural similarity with peers in the dorsomedial prefrontal cortex (DMPFC) bilaterally (left DMPFC: $\beta = 0.964$, SE $= 0.233$, $p_{corrected} = 0.012$, $p_{uncorrected} < 0.001$; right DMPFC: $\beta = 0.977$, SE $= 0.232$, $p_{corrected} = 0.012$, $p_{uncorrected} < 0.001$) and right precuneus ($\beta = 0.912$, SE $= 0.237$, $p_{corrected} = 0.020$, $p_{uncorrected} < 0.001$). We show these results in Fig. 2c. In Supplementary Fig. 2, we show scatter plots to illustrate the relationships between the in-degree centrality and mean neural similarity of participants in key brain regions. We did not find any significant associations in subcortical regions (see Supplementary Table 2). We also fit analogous models to control for demographic variables that may be associated with neural similarity[26,40], models that only incorporated neural similarities between participants who were living in

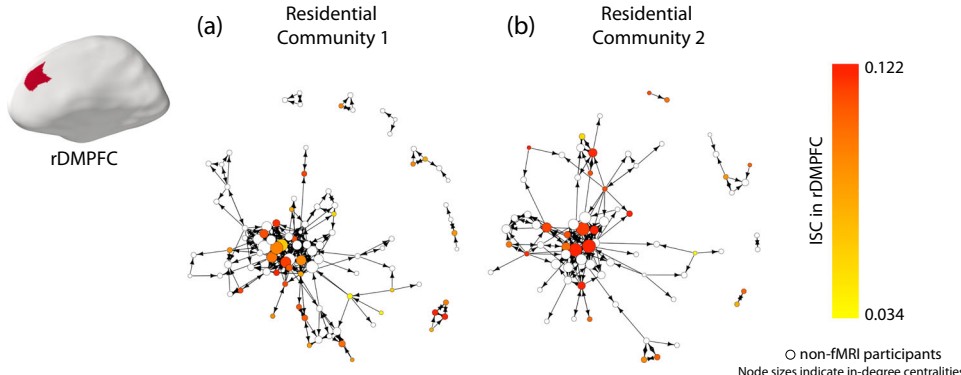

**Fig. 3 Visualization of participant-level ISC results in the social networks.** Visualizations of the social networks of (**a**) residential community 1 and (**b**) residential community 2 of a first-year dorm. Each participant was a resident of one of two distinct residential communities, where one "community" consists of the set of people who live in the same wing and floor of a residence hall. Each node (which we show as a disc) represents one resident who was living in one of the communities, and each line segment represents one directed edge between two nodes if it is unidirectional and represents two directed edges if it is bidirectional. For example, an arrow from node A to node B conveys that node A nominated node B as a friend. An edge with two arrowheads indicates a mutually nominated friendship. The size of a node represents its in-degree centrality, with larger nodes indicating individuals with higher in-degree centralities. The color of the nodes represents a node's mean neural similarity in the rDMPFC to other members of its residential community, with darker colors indicating greater neural similarities. As this figure indicates, individuals with higher in-degree centralities (i.e., individuals who many other individuals nominated as a friend) tended to have the largest mean ISCs with their peers in the rDMPFC. Source data are provided as a Source Data file.

the same residential community, models that controlled for social distances between participants in the same community, and models that used a subset of the data with approximately equal-sized centrality groups. These other approaches yielded similar results. See Supplementary Figs. 3–6.

To confirm that our results from analyzing binarized-centrality groups also hold when we treat in-degree centrality in its original (i.e., non-binarized) form, we also conducted an analogous analysis to relate participants' mean ISCs with each other in each brain region with the non-binarized in-degree centrality values. For each brain region, we calculated the Spearman rank correlation $\rho$ to examine the relationship between the mean ISCs in each brain region and in-degree centrality. We again employed FDR correction to correct for multiple comparisons across brain regions. Using these computations, we identified similar regions as when we used binarized in-degree centrality (i.e., as low versus high values). We found that neural similarities in the bilateral DMPFC (left DMPFC: $\rho = 0.420$, $p_{corrected} = 0.048$, $p_{uncorrected} < 0.001$; right DMPFC: $\rho = 0.415$, $p_{corrected} = 0.048$, $p_{uncorrected} < 0.001$), precuneus ($\rho = 0.408$, $p_{corrected} = 0.048$, $p_{uncorrected} < 0.001$), and left superior parietal lobule ($\rho = 0.424$, $p_{corrected} = 0.048$, $p_{uncorrected} = 0.002$) were significantly correlated with in-degree centrality (see Fig. 2d). In other words, we found that there was a positive association between an individual's in-degree centrality and their mean neural similarity with their peers in the DMPFC, precuneus, and superior parietal lobule. See Fig. 3 for a visualization of the ISC in the right DMPFC and its association with in-degree centrality. We did not find any significant associations in subcortical regions (see Supplementary Table 3).

Notably, for both sets of analyses, we found a positive relationship in all cases in which participants' ISCs with their peers were related significantly to their in-degree centrality. That is, in both analyses, we found that higher in-degree centralities were associated with more typical neural responses. Additionally, in these analyses and in all of our other analyses, we did not find any regions in the brain in which low in-degree centrality was associated with more similar neural responses to one's peers.

**Preference similarity.** After the neuroimaging portion of the fMRI study, participants rated the extent to which they felt that each video that they saw in the scanner was enjoyable and

interesting. For each participant, we took the following steps to calculate their mean similarity with their peers in enjoyment and interest ratings. For each of the 1952 unique dyads (i.e., pairs of individuals), we calculated the Euclidean distance between the two participants' enjoyment ratings across the 14 different videos and transformed the distance measure into a normalized similarity measure (where the similarity is given by $s = 1 − [distance/max(distance)]$). Larger similarity values, which range from 0 to 1, indicate greater similarity in how much two individuals in a dyad enjoyed the content. We repeated the same process for interest ratings. This yields two preference similarity measures per dyad.

**Analysis of participant-level preferences.** We were interested in (1) whether individuals who were highly central in their residential community had preferences that were more similar to others in their community than less-central individuals and (2) if such self-reported differences in preferences could account for the neural results that we reported above. To investigate this, we transformed the dyad-level preference similarity measures to participant-level variables. First, we calculated each participant's mean similarity in enjoyment ratings with each other participant. This estimates the extent to which each participant, on average, had similar preferences to other participants in how enjoyable they found the videos. We repeated the same process for the interest ratings. This approach yields one number for each participant to represent their mean similarity with their peers in enjoyment ratings and one number to represent their mean similarity with their peers in interest ratings. We then related the mean enjoyment and interest similarity measures with the binarized in-degree centrality variable by fitting a GLM for each similarity measure with $z$-scores of the similarity measures as the dependent variables and the binarized in-degree centrality variable as the independent variable. Our results indicate that individuals who had higher centralities in their social networks were more similar, on average, than less-central individuals with their peers in the content that they found to be enjoyable ($\beta = 0.578$, $SE = 0.253$, $p = 0.026$) and interesting ($\beta = 0.491$, $SE = 0.256$, $p = 0.061$). The association between in-degree centrality and mean interest similarity is only marginally statistically significant (i.e., trend-level).

Given our finding that individuals with a high in-degree centrality were more similar to their peers in self-reported content preferences than those with a low in-degree centrality, we tested whether our findings that link ISC to in-degree centrality could arise from inter-subject similarities in self-reported preferences. To investigate this possibility, we fit GLMs to test the relationship between the ISC in each brain region and in-degree centrality while controlling for similarity in enjoyment and interest ratings. Our results indicate that the relationships between ISC and in-degree centrality remain significant after controlling for similarity in enjoyment and interest ratings (see Supplementary Fig. 7), suggesting that neural similarity in these regions captures similarities beyond what one can attribute purely to self-reported preference ratings.

**Dyad-level ISC analysis.** Our participant-level ISC results indicate that individuals with high in-degree centralities (i.e., those who were nominated as a regular social partner by many individuals) had, on average, greater neural similarity with their peers than individuals with low in-degree centralities. We also took a finer-grained approach to test if individuals with similar in-degree centralities were most similar to one another, irrespective of whether they had a high or a low in-degree centrality, or if individuals who were highly central in their residential community were most similar to other highly-central individuals and less-central individuals were comparatively idiosyncratic (i.e., dissimilar to others, including other individuals with low in-degree centralities). To relate our dyad-level neural similarity measure with individuals' in-degree centralities, we transformed the participant-level binarized in-degree centrality measure into a dyad-level variable. We categorized the dyads into (1) {high, high} if both participants in the dyad had a high in-degree centrality, (2) {low, low} if both participants in the dyad had a low in-degree centrality, and (3) {low, high} if one participant in the dyad had a low in-degree centrality and the other participant had a high in-degree centrality. For each of our 214 brain regions, we fit a linear mixed-effects model with crossed random effects to account for the dependency structure of the data[41] (see the Methods section) with ISC in the corresponding brain region as the dependent variable and the dyad-level centrality variable as the independent variable. We then performed a planned-contrast analysis[42] to compare the different in-degree centrality groups and thereby identify brain regions for which including one or more low-centrality individuals in a dyad was associated with less-coordinated neural responses (i.e., $ISC_{\{high,high\}} > ISC_{\{low,low\}}$, $ISC_{\{high,high\}} > ISC_{\{low,high\}}$, and $ISC_{\{low,high\}} > ISC_{\{low,low\}}$) (see Fig. 4a).

The $ISC_{\{high,high\}} > ISC_{\{low,low\}}$ contrast is our most direct test of the hypotheses that highly-central individuals have very similar neural responses to one another, whereas less-central individuals have neural responses that are comparatively idiosyncratic. This is the case because it tests whether neural similarity is greater in dyads in which both individuals have high in-degree centralities than in dyads in which both individuals have low in-degree centralities. By contrast, $ISC_{\{high,high\}} > ISC_{\{low,high\}}$ would also hold for a nearest-neighbor model[33], which reflects the assumption that individuals who are more similar in a behavioral trait also exhibit greater neural similarity. (We use the term "nearest-neighbor model" to refer to the assumption that the neural responses of individuals are most similar to those of their immediate neighbors, regardless of their absolute position on some scale [31], such as whether they have a high or low in-degree centrality.) Additionally, $ISC_{\{low,high\}} > ISC_{\{low,low\}}$ does not necessarily have to hold to support the hypotheses that highly-central individuals have more similar neural responses to one another but that less-central individuals have neural responses that are comparatively idiosyncratic. For example, if each low-

centrality participant responded in a completely unique way, then they would have similarly low ISCs with other low-centrality individuals and with high-centrality individuals. Nevertheless, we reasoned that $ISC_{\{low,high\}} > ISC_{\{low,low\}}$ was likely to arise in our data set because of underlying stimulus-driven responses that are shared across all participants; each low-centrality individual will partially reflect these shared stimulus-driven responses (and they may each deviate from the typical responses in an idiosyncratic way). Accordingly, we report the results of three contrasts: (1) $ISC_{\{high,high\}} > ISC_{\{low,low\}}$, which is the most direct test of our hypotheses; (2) $ISC_{\{high,high\}} > ISC_{\{high,low\}}$, which is a test of our hypotheses but also holds for a nearest-neighbor model; and (3) $ISC_{\{low,high\}} > ISC_{\{low,low\}}$, which does not have to hold to support our hypotheses, but which we expect to hold.

We illustrate the results of the three contrasts ($ISC_{\{high,high\}} > ISC_{\{low,low\}}$, $ISC_{\{high,high\}} > ISC_{\{low,high\}}$, and $ISC_{\{low,high\}} > ISC_{\{low,low\}}$) in Fig. 4. As in our participant-level results, our dyad-level results reveal that there were larger ISCs in the DMPFC, precuneus, and portions of the superior parietal lobule in dyads of individuals who both had high in-degree centralities (i.e., {high, high}) than in dyads of individuals who both had low in-degree centralities (i.e., {low, low}) (see Fig. 4b). Additionally, ISCs in the ventrolateral prefrontal cortex (VLPFC) and temporal pole were larger in {high, high} dyads than in {low, low} dyads. ISCs in subcortical regions (including the amygdala, hippocampus, left pallidum, and right thalamus) were larger in {high, high} dyads than in {low, low} dyads (see Supplementary Table 4). We found similar patterns when we contrasted high-centrality dyads with mixed-centrality dyads ($ISC_{\{high,high\}} > ISC_{\{low,high\}}$) and mixed-centrality dyads with low-centrality dyads ($ISC_{\{low,high\}} > ISC_{\{low,low\}}$), although the effect sizes were smaller. See Fig. 4b, c and Supplementary Tables 5 and 6. In the Supplementary Information, we report results of analogous models that control for demographic variables and friendship (see Supplementary Fig. 8) and that examine neural similarities only in participants who live in the same residential community (see Supplementary Fig. 9). The latter analysis allowed us to control for both demographic similarities and social distances between individuals (see Supplementary Fig. 10). We also report the results of models that use a subset of the data with approximately equal-sized centrality groups (see Supplementary Fig. 11). The results of these additional analyses are similar to those in Fig. 4. Across all of our analyses, we did not find any regions in the brain in which there were larger ISCs in {low, low} dyads than in {high, high} dyads. We also did not find any regions in the brain in which there were larger ISCs in {low, high} dyads than in {high, high} dyads, nor any in which there were larger ISCs in {low, low} dyads than in {low, high} dyads. Our findings suggest that highly-central individuals were very similar in their neural responses, whereas less-central individuals had neural responses that were dissimilar both to highly-central individuals and to other less-central individuals. In other words, less-central individuals had neural responses that were idiosyncratic, with each less-central individual differing from the typical response of other individuals in their own way.

We also conducted an analogous analysis to relate mean ISCs with the original non-binarized values of the dyad-level in-degree centralities. To do this, we related the minimum in-degree centrality of each dyad to neural similarity in each of our 214 brain regions. Taking the minimum in-degree centrality value of each dyad allowed us to test the hypothesis that only dyads with two highly-central individuals had very similar neural responses to one another. If a low in-degree centrality is associated with an idiosyncratic neural response, then the inclusion of even just one low-centrality individual in a dyad should be associated with a

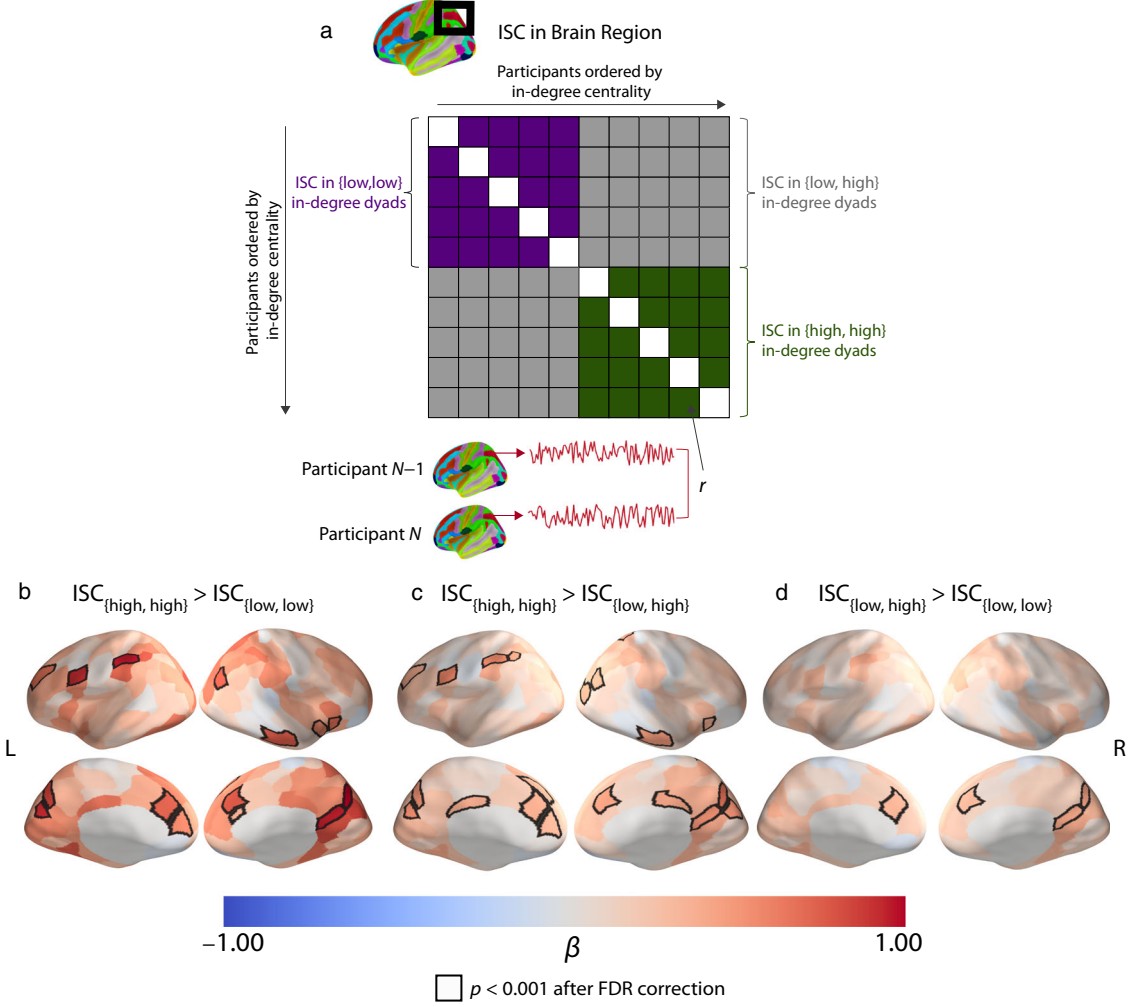

**Fig. 4 Dyad-level analysis and results. a** Dyad-level ISCs in a brain region are encoded in a matrix whose entries consist of pairwise Pearson correlation coefficients. The rows and columns of the matrix are ordered according to the in-degree centralities of the participants. We performed planned contrasts of the different centrality groups to test whether dyads in which both individuals were highly central (i.e., ISC$_{\{high,high\}}$), had larger ISCs than dyads in which both individuals were less central (i.e., ISC$_{\{low,low\}}$) and than dyads with mixed centralities (i.e., ISC$_{\{low,high\}}$), for which one individual of the dyad had a low centrality and the other had a high centrality. [The figure in (**a**) is adapted from prior work[34].] **b** There were larger ISCs in the DMPFC, VMPFC, VLPFC, precuneus, temporal pole, and portions of the superior parietal lobule in {high, high} dyads than in {low, low} dyads. **c** We found similar patterns when we compared {high, high} dyads to {low, high} dyads and **d** when we compared {low, high} dyads to {low, low} dyads. The ISC$_{\{high,high\}}$ > ISC$_{\{low,low\}}$ contrast in (**b**) provides the most direct test of our main hypotheses that highly-central individuals have similar neural responses to one another and that less-central individuals have neural responses that are idiosyncratic. The quantity β is the standardized regression coefficient. Regions with significant differences for each contrast are outlined in black. We used an FDR-corrected significance threshold of $p < 0.001$, which corresponds to an uncorrected $p$-value threshold of $p < 6.386 \times 10^{-5}$. All of the reported $p$-values are two-tailed. Source data are provided as a Source Data file.

small ISC. For each brain region, we fit a linear mixed-effects model with crossed random effects to account for the dependency structure of the data[41] (see the Methods section) with the ISC in the corresponding brain region as the dependent variable and the log-transformed minimum in-degree centrality value [specifically, we used $\ln(1 + \text{minimum in-degree centrality})$] of each dyad as the independent variable.

As with our dyad-level results using the binarized centrality variable, we found a positive association between the minimum in-degree centrality of the dyads and neural similarity in the left DMPFC, precuneus, posterior cingulate cortex, superior parietal lobule, and middle temporal gyrus. That is, there was greater neural similarity in these brain regions in dyads with a higher minimum in-degree centrality. Mirroring our results with a binarized in-degree centrality variable, dyads in which both individuals were highly central in their residential community (as encoded by a high minimum in-degree centrality) had greater

neural similarity than dyads in which both individuals were less central (as encoded by a low minimum in-degree centrality) (see Fig. 5).

**Analysis of dyad-level preferences.** We tested whether the self-reported preferences of individuals were consistent with the hypotheses that more-central individuals have preferences that are very similar to one another and that less-central individuals have preferences that are idiosyncratic, with each low-centrality individual's preferences differing from those of other individuals in their own way. We also tested if such self-reported differences in preferences could account for our neural results. We first fit two mixed-effects models, with crossed random effects to account for the dependency structure of the data[41]. (See the Methods section.) We employed one such model for each type of preference (i.e., similarities in enjoyment and interest ratings). We used dyad-level similarities in enjoyment and interest ratings

Dyad-level results: Associating neural similarity
with the minimum in-degree centrality in dyads

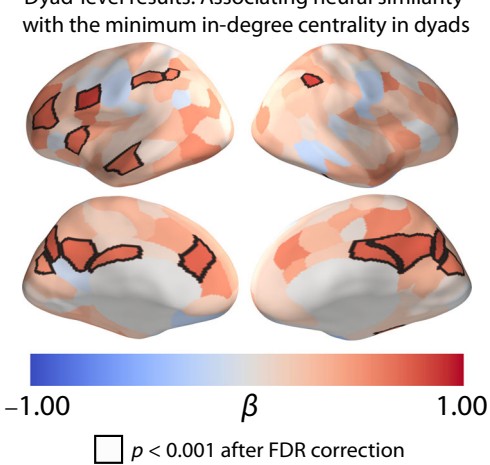

−1.00          $\beta$          1.00

☐ $p < 0.001$ after FDR correction

**Fig. 5 Dyad-level associations of neural similarity with the minimum in-degree centrality of dyads.** We found a positive association between ISC and minimum in-degree centrality. Larger ISCs in brain regions (including the DMPFC, the VLPFC, the precuneus, the temporal pole, and portions of the superior parietal lobule) were associated with a higher minimum in-degree centrality. The quantity β is the standardized regression coefficient. Regions where we observed significant associations between in-degree centrality and ISC are outlined in black. We used an FDR-corrected significance threshold of $p < 0.001$, which corresponds to an uncorrected $p$-value threshold of $p < 8.879 \times 10^{-5}$. All of the reported $p$-values are two-tailed. Source data are provided as a Source Data file.

(see the above discussion of preference similarity) as the dependent variables—one in each of the two models—and the dyad-level minimum-centrality variable as the independent variable. We then performed planned contrasts of the three different dyad-level centrality groups (i.e., {high, high}, {low, low}, and {low, high}) to test if the inclusion of one or more low-centrality individuals in a dyad was associated with lower levels of interpersonal similarities in preferences (i.e., $s_{\{high,high\}} > s_{\{low,low\}}$, $s_{\{high,high\}} > s_{\{low,high\}}$, and $s_{\{low,high\}} > s_{\{low,low\}}$, where $s$ corresponds to dyad-level preference similarity, as defined in the above discussion of preference similarity). We employed FDR correction to correct for multiple comparisons from the multiple planned contrasts. Our results indicate that dyads that consisted of two highly-central individuals (i.e., {high, high}) were more similar to one another in what they found enjoyable and interesting than dyads that consisted of two less-central individuals (i.e., {low, low}) (see Supplementary Tables 7 and 8). We found similar patterns when we compared high-centrality dyads to mixed-centrality dyads and when we compared mixed-centrality dyads to low-centrality dyads (see Supplementary Tables 7 and 8).

We then tested whether the above behavioral findings, which (like our neural findings) support the hypotheses that highly-central individuals are similar in their preferences and that less-central individuals have idiosyncratic preferences, could account for our neural results. Specifically, we examined whether inter-subject similarities in self-reported preferences could explain our observation that individuals who were highly central in their residential community were similar in their neural responses to other highly-central individuals and that less-central individuals were comparatively idiosyncratic. To examine this possibility, we fit additional linear mixed-effects models to test the relationship between ISCs in each brain region and dyad-level in-degree centrality (i.e., whether a given dyad was composed of two high-centrality individuals, two low-centrality individuals, or one high-centrality individual and one low-centrality individual) while controlling for similarity in enjoyment and interest ratings.

Although similarities in enjoyment and interest ratings were also associated with neural similarity in regions of the default mode network (see Supplementary Figs. 11 and 12), our results indicate that the Anna Karenina pattern of results that links ISCs and dyad-level in-degree centralities remains significant after controlling for similarity in enjoyment and interest ratings (see Supplementary Fig. 14). Therefore, we conclude that our findings that greater neural similarity tended to occur between highly-central individuals and that comparatively low neural similarity tended to occur between less-central individuals arose from differences beyond those that were captured by self-reported preference ratings.

## Discussion

What factors distinguish highly-central individuals in social networks? Our results are consistent with the notion that individuals who are central in their social networks process external stimuli in typical ways, whereas less-central individuals process external stimuli idiosyncratically. In our study, we found that highly-central individuals exhibited greater mean neural similarity with their peers than less-central individuals in several regions of the brain, including regions in which similar neural responding has been associated with shared high-level interpretations of events and social cognition (e.g., regions of the default mode network) while viewing dynamic, naturalistic stimuli[43]. We observed a distinct pattern in the relationship between centrality and neural similarity: highly-central individuals were very similar to one another in their neural responses, whereas less-central individuals were dissimilar both to one another and to their peers' typical ways of processing external stimuli. Our findings are consistent with the possibility that highly-central people process and respond to external stimuli in a manner that allows them to relate to and connect with many of their peers and with the possibility that less-central people have idiosyncrasies that may result in greater difficulty in relating to others. However, longitudinal research is needed to determine if social-network centrality causes or results from processing external stimuli in a way that is similar to one's peers.

Brain areas in which highly-central individuals exhibited, on average, greater neural similarity with their peers than was the case for less-central individuals included the bilateral DMPFC and precuneus, which are both regions of the default mode network. Mirroring our findings that link in-degree centrality and mean neural similarity with community members, brain areas in which highly-central individuals responded similarly to one another and less-central individuals responded idiosyncratically include the DMPFC, precuneus, and other regions of the default mode network (such as the posterior cingulate cortex and the inferior parietal lobule). These regions have been implicated in social cognitive processes such as mentalizing and perspective-taking[44–46]. Neural similarity in these regions has also been associated with similarities in the understanding and interpretation of narratives, presumably because people who share similar viewpoints and perspectives have greater similarity in these higher-order brain regions during the viewing of naturalistic stimuli than those who do not[29,34,47]. Additionally, neural similarity in these regions has been associated with friendship[26]; friends (i.e., people who report regularly socializing with each other) appear to have greater similarity in these regions than people who are not friends. In particular, it was suggested recently[43] that the default mode network helps promote a critical sense-making function by combining external information about one's surroundings with internal experiences and schemas to create models of situations as they unfold over time and that ISCs in regions of the default mode network support the creation of

shared meaning across individuals. Our results suggest that highly-central individuals process external stimuli in a way that closely reflects their peers' typical ways of understanding and responding to external stimuli. Such similarity may help them relate and connect to many people (although further work is needed to elucidate the causal mechanisms that underlie our results). Our findings also suggest that highly-central individuals are similar to one another and that less-central individuals are dissimilar from a group's typical ways of processing and understanding external stimuli (such that they each process and respond to external stimuli in their own idiosyncratic way). Our results were significant even when we controlled for (1) demographic variables that may be associated with neural similarity and (2) social distances between individuals. Therefore, our findings suggest that the association of neural similarity in regions of the default mode network (and in other regions) with in-degree centrality is not merely a result of the most-central individuals being more likely to be friends with one another. Instead, we observed that highly-central individuals had very similar neural responses to many of their peers, including those with whom they were not friends.

In our study, highly-central individuals also self-reported preferences for the stimuli that were more reflective of their peers' preferences. Specifically, highly-central individuals had greater mean similarity with their peers in the extent to which they found stimuli to be enjoyable and interesting. Furthermore, highly-central individuals had similar preferences for the stimuli as one another, but each less-central individual had idiosyncratic preferences for the stimuli (i.e., preferences that were different both from the preferences of their peers on average and from those of other less-central individuals). In concert, the observed behavioral patterns suggest that highly-central individuals self-report preferences that are more aligned with their peers' preferences and thus may be more in tune with what others find enjoyable or interesting; this may help them connect with their peers through mutually shared interests. Notably, controlling for similarities in the enjoyment and interest ratings did not change our results that link neural similarity with in-degree centrality in social networks. That is, we found that neural similarity in brain regions that have been implicated in high-level interpretation and social cognition was associated with network centrality above and beyond what we were able to capture using self-reported preferences. This suggests that measuring neural responses to naturalistic stimuli as they unfold over time can capture consequential aspects of mental processing beyond what one can obtain using a few targeted self-report questions. The strong link between in-degree centrality and ISCs (even when controlling for similarities in participants' self-reported preferences), relative to links between in-degree centrality and similarities in self-reported preferences, may arise from several factors, including the finer temporal granularity of ISCs than our self-report measures (because ISCs capture similarities in how responses evolve with time), the limits of self-report (because people are often unaware of and/or unwilling to report features of their attitudes and other aspects of their mental processing[48]), and the possibility that the similarities in processing that are linked to centrality reflect similarities in the creation of internal models of situations as they evolve with time (rather than reflecting similarities in what participants found interesting or enjoyable)[43]. A notable benefit of calculating ISCs is that one can use them to characterize similarities in many different aspects of mental processing in parallel. One can thereby obtain insight into diverse emotional and cognitive processes that unfold in response to various situations and that may be shaped by individuals' pre-existing beliefs, values, attitudes, and experiences. Our neural findings also provide insights that can inform which self-report measures are likely to capture the types of processing

that may be particularly similar between highly-central individuals and their peers. It may be particularly fruitful to test for associations between in-degree centrality and the typicality of (1) responses to self-report scales that capture individuals' social and cognitive tendencies and/or (2) individuals' interpretations of stimuli (e.g., as captured by semantic analyses of free-response measures).

In the present study, we obtained data from two different residential communities and characterized the neural similarity of each participant in our study with each other participant, including individuals from the other community. Furthermore, we successfully replicated all of our main effects that link ISCs and in-degree centrality when we fit models using only intra-community dyads. (See Supplementary Figs. 4 and 9.) The two residential communities each consisted of first-year students who were attending the same university. Although each residential community is relatively bounded and interactions between community members were likely to be uncommon—both because of restrictions that arose from the building structure and because of programming that focused on intra-community social activities—it is likely that the two residential communities had similar shared values, beliefs, and experiences. Therefore, the types of similarities in processing external stimuli that are associated with in-degree centrality in one residential community are likely to be similar to those that are associated with in-degree centrality in the other community. However, in some contexts, looking specifically at only intra-community similarities in neural activity may be important when relating ISCs with social-network centralities, particularly when drawing on participants from communities with norms that are markedly different from each other. Future work can further elucidate the extent to which ISCs within and between communities are associated with individual differences in the centralities of individuals in social networks.

Because our study has only one wave of fMRI data, an important limitation of our work is that we are not able to ascertain the causal mechanisms that drive our effects. Additional research is necessary to disentangle various possible causes. We hope that such research will help discern whether (1) processing external stimuli in similar ways to their peers causes certain individuals to become highly central in their social networks, (2) being highly central in a social network causes certain individuals to process external stimuli in ways that are more similar to those of their peers, or (3) some combination of these two possibilities is at play. Moreover, if being highly central causes people to process stimuli similarly to their peers, future research can also help uncover whether (1) highly-central individuals (as a result of their central positions in a network) exert influence on others in their social network, so that many individuals in the network become more similar to the highly-central individuals, (2) highly-central individuals change the way that they process external stimuli to fit the norms of a social network, or (3) some combination of these two possibilities is at play. Research that uses longitudinal designs will be important for arbitrating between these possibilities.

Prior research suggests that individuals with high weighted in-degree centralities have more behavioral and neural sensitivity to interpersonal cues than individuals with low weighted in-degree centralities[49] and that such highly-central individuals are more likely than less-central individuals to adapt their brain activity to match that of other individuals in their social group[50]. Therefore, one possibility is that people who are highly central may adapt their views to meet their social network's typical ways of processing the world, perhaps due to a greater need to belong socially or a desire to connect with a large number of people. Future studies that employ longitudinal data can help elucidate the direction(s) of these effects and further clarify the mechanisms that may be at play. For instance, pairing in-lab measures of social

conformity with longitudinal studies that examine neural similarity may provide insight into the extent to which individual differences in in-lab measures of conformity are predictive of changes in individuals' unconstrained processing of external stimuli. Another possibility is that highly-central individuals may show similarly high levels of social abilities and functioning, which may in turn lead to greater neural similarity with others. For instance, it is possible that individuals with high in-degree centrality may have distinctively high levels of empathic concern, mentalizing abilities, and/or emotion-regulation abilities that help them form and maintain a large number of social ties and also impact how they respond to naturalistic stimuli. Therefore, potential differences in social functioning between individuals with high in-degree centralities and other individuals may be a key reason for the links between in-degree centrality and neural similarity that we found in the present study. We did not collect measures that can capture individual differences in social functioning, so we are not able to test these theories using our data. Future studies that investigate associations between individual differences in social functioning, centrality, and neural processing of naturalistic stimuli can further elucidate these relationships.

In summary, our results suggest that highly-central individuals in social networks are very similar to their peers in how they process external stimuli, as indicated by neural responses that are associated with social cognition and building shared internal models of situations. We also found support for the idea that highly-central individuals are similar to one another in their neural processing but that less-central individuals are each dissimilar from their peers and from one another in their own idiosyncratic ways. Overall, our results suggest that a similar understanding of the world, as reflected in similar brain responses across people, may help humans achieve and maintain social connections.

## Methods

**Characterization of the social networks in the two residential communities**. A total of 119 participants completed our social-network survey, with $n_{residential\ community\ 1} = 70$ and $n_{residential\ community\ 2} = 49$ people in the two residential communities. All participants were living in one of these two communities of a first-year dormitory in a large state university (University of California, Los Angeles) in the United States. All participants provided informed consent for the social-network survey in accordance with the Institutional Review Board of the University of California, Los Angeles. The survey was administered during December and January of the students' first year in the university, which began in the last week of September. Therefore, the participants had been living together in their communities for 3–4 months prior to completing the social-network survey. The participants were compensated with $15 for completing the survey. In the survey, the participants were first asked to indicate their full names and any nicknames by which they were known. This allowed us to match individuals' names with the number of nominations that they received from other residents of their community. The participants were then asked to type the names of other people in their residential community with whom they interacted regularly. They answered the following prompt: "Consider the people you like to spend your free time with. Since you arrived at [institution name], who are the people you've socialized with most often? (Examples: eat meals with, hang out with, study with, spend time with)." The participants in the study could name as many people as they wished who fit that description without any restrictions, and no time limit was imposed on the survey. We adapted this question from prior research that investigated social networks of university students[13,26,51].

We used the IGRAPH package 1.2.4[52] in R 3.6.1[53] to analyze the social-network data. We constructed two networks (i.e., one for each residential community) and encoded the participants' answers with unweighted and directed edges. We then calculated the in-degree centrality of each individual. This quantity gives the number of the individual's community members (who participated in the social-network survey) who named them as someone with whom they interacted regularly. The distributions of the in-degree centralities were similar across the subsets of the fMRI sample from each community (see Supplementary Fig. 1).

**Combining data across residential communities**. As we noted in the prior subsection, each participant in our study was living in one of two residential communities and we calculated in-degree centrality as the number of nominations that each participant received from peers who were living in the same residential community. Each residential community was relatively bounded, and residents

were encouraged (e.g., via intra-community social activities) to form social connections within their community. To maximize statistical power, we compared the neural responses across all possible pairs of participants (i.e., dyads) in both residential communities and then related the ISCs to in-degree centrality values across all possible pairs of individuals, including ones who were living in different residential communities. It is possible that this approach may have diminished our capacity to detect relationships between neural similarity and in-degree centrality, depending on how much the link between in-degree centrality within communities and neural similarity is based on community-specific norms. However, both communities consisted of first-year students who were attending the same university, so we reasoned that the two communities were likely to have similar shared values, beliefs, and experiences and that it would thus be reasonable for our neural analysis to include ISCs between individuals from different residential communities. We later complemented these main analyses with analyses that were based on only intra-community neural similarities. The results of these subsequent analyses (see Supplementary Figs. 4, 5, 9, and 10) yield similar results as our main analyses.

**fMRI study participants**. A total of 70 participants from the aforementioned two residential communities participated in the neuroimaging portion of our study. We excluded two individuals due to excessive movement in more than half of the scan and excluded one individual who fell asleep during half of the scan. We also excluded one individual who did not complete either the scan or the social-network survey. Three additional fMRI participants did not complete the social-network survey. (See Supplementary Table 9 for a table of excluded participants.) This resulted in a total of 63 participants (40 female) between the ages of 18 and 21 (with a mean age of $M = 18.19$ and a standard deviation of SD = 0.59) that we included for all analyses. The distributions of in-degree centralities were similar across the fMRI participants and the full set of participants (see Supplementary Fig. 1). We included partial data from two fMRI participants. One participant had excessive head movement in one of the four runs, and one participant reported falling asleep in one of the four runs. In analyses that involved brain data, we excluded the associated runs for these participants and only included the remaining three runs. All participants provided informed consent in accordance with the procedures of the Institutional Review Board of the University of California, Los Angeles.

**fMRI procedure**. Participants attended an in-person study session that included self-report surveys and a 90-minute neuroimaging session in which we measured their brain activity using blood-oxygen-level-dependent (BOLD) fMRI. The fMRI data collection occurred between September and early November during the participants' first year at the university, and it was thus completed before the start of data collection for the social-network part of our study. Prior to entering the scanner, participants completed self-report surveys in which they provided demographic information, including their age, gender, and ethnicity. During the fMRI portion of the study, the participants watched 14 video clips with sound. The stimuli consisted of 14 different videos that varied in both duration (from 91 to 734 seconds) and content. (See Supplementary Table 1 for descriptions of the content.) Prior to scanning, we informed the participants that they would be watching video clips of heterogeneous content and that their experience would be like watching television while someone else "channel-surfed". (The term "channel-surfing" is an idiom that refers to scanning through different television channels to find something to watch.) The video clips were presented across four runs (as described in Supplementary Table 1) without breaks between clips within each run. The participants were paid $50 for completing the fMRI study.

Some of the video clips have been used previously (10 of the videos were used in prior studies, so 4 of them are new), and we used similar criteria to those in prior work to select new stimuli[26,40]. First, we selected stimuli that were unlikely to have been seen previously by the participants in an effort to avoid inducing inter-subject differences that could arise from familiarity with the content. Second, we selected stimuli that were likely to be engaging to minimize the likelihood that participants would mind-wander during viewing, as this could potentially introduce undesirable noise into our data. Third, we selected stimuli that were likely to elicit substantial variability in the interpretations and meaning that different individuals can draw from the content. The participants were asked to watch the videos naturally (i.e., as they would watch them in a normal situation in life). All participants saw the videos in the same order to avoid any potential variability in neural responses from differences in the way that the stimuli were presented (rather than from endogenous individual-level differences). One can think of our consistently ordered series of stimuli as a single continuous stream of content (analogous to different scenes in a movie). It is possible that different relative orderings of the stimuli could generate different results, similar to how reordering scenes in a movie might generate different results (e.g., due to differences in how tone, narratives, and themes evolve over the span of a movie). In our study, we presented the videos in the same order to all participants to keep the context surrounding each video consistent across participants because our main priority was to maximize sensitivity to individual-level differences. The video task was divided into four runs, and the task lasted approximately 60 minutes in total. Structural images of the brain were also collected. (We describe the image collection in more detail in the

subsection on fMRI data acquisition.) After the fMRI scan, the participants provided ratings (in the form of integers between 1 and 5) both on how much they enjoyed each video ("How much did you enjoy this video?"; response options ranged from 1 to 5, with the anchors "1 = not at all" and "5 = very much") and on how interesting they found each video ("How interesting did you find this video?"; response options ranged from 1 to 5, with the anchors "1 = very boring" and "5 = very interesting"). We obtained these preference ratings after the fMRI scan in an effort to minimize potential biases or disruptions in processing that could occur if participants were asked to reflect on content immediately after each stimulus was presented during scanning.

**fMRI data acquisition**. The participants were scanned using a 3 T Siemens Prisma scanner with a 32-channel coil. Functional images were recorded using an echo-planar sequence (with echo time = 37 ms, repetition time = 800 ms, voxel size = 2.0 mm × 2.0 mm × 2.0 mm, matrix size = 104 × 104 mm, field of view = 208 mm, slice thickness = 2.0 mm, multi-band acceleration factor = 8, and 72 interleaved slices with no gap). A black screen was included at the beginning (with duration = 8 s) and the end (duration = 20 s) of each run to allow the BOLD signal to stabilize. We also acquired high-resolution T1-weighted (T1w) images (with echo time = 2.48 ms, repetition time = 1900 ms, voxel size = 1.0 mm × 1.0 mm × 1.00 mm, matrix size = 256 × 256 mm, field of view = 256 mm, slice thickness = 1.0 mm, and 208 interleaved slices with a 0.5 mm gap) for coregistration and normalization. We attached adhesive tape to the head coil in the MRI scanner and applied it across the participants' foreheads; it is known that this significantly reduces head motion[54].

**fMRI data analysis**. We used fMRIPrep version 1.4.0 for the data processing of our fMRI data[55]. We have taken the descriptions of anatomical and functional data preprocessing that begins in the next paragraph from the recommended boilerplate text that is generated by fMRIPrep and released under a CC0 license, with the intention that researchers reuse the text to facilitate clear and consistent descriptions of preprocessing steps, thereby enhancing the reproducibility of studies.

For each participant, the T1-weighted (T1w) image was corrected for intensity non-uniformity (INU) with N4BiasFieldCorrection, distributed with ANTs 2.1.0[56], and used as T1w-reference throughout the workflow. Brain tissue segmentation of cerebrospinal fluid (CSF), white matter (WM), and gray matter (GM) was performed on the brain-extracted T1w using FSL fast[57]. Volume-based spatial normalization to the ICBM 152 Nonlinear Asymmetrical template version 2009c (MNI152NLin2009cAsym) was performed through nonlinear registration with antsRegistration (ANTs 2.1.0[56]).

For each of the four BOLD runs per participant, the following preprocessing was performed. First, a reference volume and its skull-stripped version were generated using a custom methodology of fMRIPrep. The BOLD reference was then coregistered to the T1w reference using FSL flirt[57] with the boundary-based registration cost function. The coregistration was configured with nine degrees of freedom to account for distortions remaining in the BOLD reference. Head-motion parameters with respect to the BOLD reference (transformation matrices, and six corresponding rotation and translation parameters) were estimated before any spatiotemporal filtering using FSL mcflirt[57]. Automatic removal of motion artifacts using independent component analysis (ICA–AROMA) was performed on the preprocessed BOLD on MNI-space time series after removal of non-steady-state volumes and spatial smoothing with an isotropic, Gaussian kernel of 6 mm FWHM (full-width half-maximum). The BOLD time series were then resampled to the MNI152Nlin2009cAsym standard space.

The following 10 confounding variables generated by fMRIPrep were included as nuisance regressors: global signals extracted from within the CBF, white matter, and whole-brain masks, framewise displacement, three translational motion parameters, and three rotational motion parameters.

**Cortical parcellation into brain regions**. We extracted neural responses across the whole brain using the 200-parcel cortical parcellation scheme of Schaefer et al.[37] and 14 subcortical regions using the Harvard–Oxford subcortical atlas[38]. Together, this resulted in 214 regions.

**Inter-subject correlations**. We used the SciPy 1.5.3 library in Python 3.7.0 to calculate ISCs. We extracted and concatenated preprocessed time-series data across all four runs for each participant, except for the two participants for whom we used only partial data. For these two participants, we concatenated their three usable runs into a single time series and then calculated ISCs for these participants by comparing their data to the corresponding three runs in the other participants. We extracted the mean time series in each of the 214 brain regions for each participant at each time point [i.e., at each repetition time (TR)]. Our analyses included 63 participants after the various exclusions, so there were 1952 unique dyads. For each unique dyad, we calculated the Pearson correlation between the mean time series of the neural response in each of the 214 brain regions. We then Fisher z-transformed the Pearson correlations and normalized the subsequent values (i.e., using z-scores) within each brain region.

**Participant-level analysis**. As we explained in the Results section, we were interested in whether an individual's in-degree centrality is associated with their mean neural similarity with their peers. To test this relationship, we transformed the dyad-level neural similarity measures into individual-level measures to obtain a single number that encoded an individual's mean neural similarity with other individuals for each brain region. For each individual, we calculated the mean Fisher z-transformed ISC value for them with each other individual in each brain region. We then fit a separate GLM for each brain region to test the association between individual differences in in-degree centrality and the mean neural similarity in the respective brain region. We FDR-corrected all results because of the multiple comparisons.

**Dyad-level ISC analysis**. For our dyad-level analysis, we took the following steps to test for associations between in-degree centrality and neural similarity in each of the 214 brain regions. First, we transformed the participant-level in-degree centrality measure into a dyad-level measure by creating a binarized variable that indicated whether the two members of the dyad had high, low, or mixed in-degree centralities (i.e., {high, high}, {low, low}, or {low, high}). See the Results section for details. Of the 1952 unique dyads, 253 of them were {high, high}, 779 of them were {low, low}, and 920 of them were {low, high}. To relate this dyad-level in-degree centrality measure and neural similarity, we used the method in Chen et al.[41] and fit linear mixed-effects models with crossed random effects using LME4 1.1-23[58] and LMERTEST 3.1.0[58] in R. This approach allowed us to account for non-independence in the data from repeated observations for each participant (i.e., because each participant is in multiple dyads). Following the method that was suggested by Chen et al.[41], we doubled the data (adding redundancy) to allow fully crossed random effects. In other words, we accounted for the symmetric nature of the ISC matrix and the fact that one participant contributes twice in a dyad (i.e., $(i, j) = (j, i)$ for participants $i$ and $j$). See Chen et al.[41] for more details. Following Chen et al.[41], we manually corrected the degrees of freedom to $N – k$, where $N$ is the number of unique observations (in our case, $N = 1952$ because there are 1952 unique dyads) and $k$ is the number of fixed effects in the model, before performing statistical inference. All findings that we report in the present paper use the corrected number of degrees of freedom. For each brain region, we first fit a mixed-effects model to infer neural similarity in that brain region from the binarized dyad-level in-degree variable, with random intercepts for each member of the dyad (i.e., participant 1 and participant 2). We then conducted planned-contrast analyses using EMMEANS[59] in R to compare which brain regions had larger ISCs for the different values of the dyadic in-degree centrality variable: $ISC_{\{high,high\}} > ISC_{\{low,low\}}$, $ISC_{\{high,high\}} > ISC_{\{low,high\}}$, and $ISC_{\{low,high\}} > ISC_{\{low,low\}}$. We transformed all variables into z-scores prior to our subsequent computations to obtain standardized coefficients ($\beta$) as outputs. We FDR-corrected all p-values at $p < 0.001$ because of multiple comparisons.

**Dyad-level behavioral analysis**. We took an analogous approach as in our dyad-level ISC analysis to test the relationships between dyadic in-degree centrality and preference similarity. (See our discussion of preference similarity in the Results section.) To do this, we followed the same procedure as the one that we described above in our discussion of dyad-level ISC analysis and fit two mixed-effects models that take into account the dependency structure of the data. We constructed one such model for each type of rating (i.e., enjoyment and interest) to infer a similarity from the dyad-level in-degree variable, with random intercepts for each member of the dyad. We then conducted planned-contrast analyses using EMMEANS 1.4.3.01[59] in R to examine whether or not there was an association between preference similarity and different levels of the dyadic in-degree centrality variable: $s_{\{high,high\}} > s_{\{low,low\}}$, $s_{\{high,high\}} > s_{\{low,high\}}$, and $s_{\{low,high\}} > s_{\{low,low\}}$, where $s$ is the dyad-level preference similarity that we defined in our discussion of preference similarity. We transformed all variables into z-scores prior to our subsequent calculations to obtain standardized coefficients ($\beta$) as outputs. We used an FDR-corrected significance threshold of $p < 0.001$ because of the multiple comparisons from the planned contrasts.

**Reporting summary**. Further information on research design is available in the Nature Research Reporting Summary that is linked to this article.

## Data availability

Source data for the figures (specifically, the inter-subject correlation and in-degree centrality data that we generated in this study) are provided with the paper and also at Zenodo at https://doi.org/10.5281/zenodo.5965852.

## Code availability

The code that we used for our analyses is available at Zenodo at https://doi.org/10.5281/zenodo.5711372.

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

## Acknowledgements

We thank Elena Sternlicht, Kelly Xue, and the UCLA Center for Cognitive Neuroscience (particularly Jared Gilbert) for providing support with data collection. We thank Oshton Tsen for providing support with the figures. This work was supported by the National Science Foundation SBE Postdoctoral Research Fellowship [Grant No. 1911783 to ECB], the National Science Foundation [Grant No. SBE-1835239 to CP and MAP], and the National Institute of Mental Health [Grant No. R01MH128720 to CP]. Figure 1 uses icons that were made by Freepik, monkik, and Becris from www.flaticon.com. Figure 2 uses icons that were made by Becris from www.flaticon.com.

## Author contributions

E.C.B., R.H., M.A.P., and C.P. designed the study and experiments. E.C.B., R.H., and K.L. collected the data. E.C.B. analyzed the data with support from R.H., E.S.F., and C.P. E.C.B., M.A.P., and C.P. wrote the manuscript with feedback from all authors.

## Competing interests

The authors declare no competing interests.

## Additional information

**Supplementary information** The online version contains supplementary material, which is available at https://doi.org/10.1038/s41467-022-28432-3.

