## [Peer Review File · Nature Communications]

In-degree centrality in a social network is linked to coordinated neural activityREVIEWER COMMENTS

Reviewer #1 (Remarks to the Author):

The authors tested whether popular individuals process the world in an exceptionally similar manner to their community peers. They found that indeed the answer is "yes", especially in regions within the default mode network. These results support the paraphrase on Lev Tolstoy's opening sentence from *Anna Karenina*, by suggesting that all popular individuals are alike, whereas unpopular individuals are unique in their own way.

I think this is an intriguing study, that provoke thoughts about what it means to be popular, from a neuroscience perspective.

Nevertheless, I do have some concerns and suggestions regarding the analysis.

Concerns:

1. The authors wanted to take a finer grain approach to test the Anna Karenina principle, and used dyad-level ISC analysis. I think this is a good approach to take. That said, I wonder why the authors divided the dyads into three groups (high-high, low-low and high-low), instead of using IS-RSA as suggested in one of the co-authors' paper (Finn, et al. *NeuroImage*, 2020)? I think it is a more sensitive and appropriate method to use for testing this.

2. It seems that the distribution of the popular participants was not uniform, and most of them had 3-4 in-degree centrality. Was the similar-to-their peers effect stronger in the seven individuals that had 10 or more In-degree centrality?

3. It was not clear to me from the analysis description – were there regions in which the low centrality individuals were more synchronized than the high centrality individuals?

4. If I understand Figure 3 correctly, it seems that the effect was much more pronounced in Community 2 than Community 1. Is this the case? And if so, do the authors have an explanation for this?

5. The authors performed an important analysis that found that individuals' enjoyment and interest from the clips did not explain the popularity effect on the neural synchronization. However, it could still be that similar interpretation of the clips (which is not reflected in enjoyment and interest) was what drove the increased neural synchronization of popular individuals. How can the authors rule out this possibility?

6. The authors used 14 video clips, which they chose in order to elicit meaningful variability in the interpretations and meaning that different individuals can draw from the content. Were there specific contents that more powerfully generated the "popularity neural synchronization effect"?

7. Did the authors test if popular individuals were more similar in their brain responses to the individuals that nominated them than to other individuals? In a way, this could be used as a replication to Parkinson et al 2018 paper on "similar neural responses predict friendship".

8. Did the authors measured participants' conformity? It could be interesting to relate one's conformity to being similar to the group's norm neural response.

Reviewer #2 (Remarks to the Author):

This manuscript builds on this research groups previous work aiming to understand the ways in which neural similarity is related to the construction of social networks. The present manuscript tests the hypothesis that people who are central in their social networks (i.e., popular) process the world similarly. The main results support this hypothesis, with highly central individuals showing high correlations during naturalistic viewing in regions comprising the default mode network, whereas those who were less central demonstrated more unique patterns of responding during naturalistic viewing.

I very much like this paper and its approach, and I think it is a logical next step from some of the senior author's previous work. However, I do have a few questions/comments that I think should

be considered before recommending publication.

1. The main question being investigated here is whether people in central positions within social networks--i.e., those who are well-connected to many others in the network--process naturalistic stimuli in was similar to their peers as compared to people who are less-central in social networks. It is stated that since 'personality traits' (i.e., extraversion, emotional stability) are inconsistently associated with being well-connected, the role of centrality within social networks may be more important to understanding and connecting with others. Yet, could there be individual differences in other characteristics that could serve as the underlying mechanism here driving the similarity in neural processing of naturalistic stimuli. For example, is it just that they are central in the network? Or is it that they all have similar levels of social abilities/function. Do the authors have any additional data on anything like mentalizing, empathy, interpersonal emotion regulation or the like that might help tease apart or further explain what is driving this network centrality effect? Again, I really like this study, but there just feels like there is some piece that is missing here.
2. The authors collected ratings of enjoyment and interest in the movie clips that participants viewed, and report similar patterns of results in terms of network centrality and subjective perceptions of the videos. They also used these ratings in a GLM to see whether the effect of network centrality held when accounting for the variance contributed by these ratings. However, I think it might also be useful to know whether these ratings indeed predicted neural similarity in the same regions (i.e., DMN) or others? Or not at all?
3. Relatedly, I was just curious about the decision to hold all ratings of enjoyment and interest in the movie clips until the end of the task, as opposed to querying participants right after the presentation of each stimulus?
4. The authors report a final N of 63 for the fMRI component of their study after exclusions, initially starting at 70. However, in reading on p28 (starting on line 566) details regarding subject exclusions, the numbers do not seem to easily add up to a total of 7 participants excluded. Can the authors clarify?
5. In the section on 'Subject-level preference analysis' (p13-14), the authors note that people who were more popular were more similar than less popular people in what they found to be enjoyable and interesting; the result for what they found 'interesting' is trend-level, and this should be acknowledged.
6. The authors presented the video clips in the same order for all participants to 'avoid any potential variability in neural responses from differences in the way that the stimuli were presented'. Yet, I think the argument could be made that without multiple orders, it is difficult to know whether the results are due solely to the way in which the stimuli appeared for everyone. I would think this should also be acknowledged as a potential limitation, even though it was intended to be an advantageous design (and it may well be).

Reviewer #3 (Remarks to the Author):

The authors had 120 participants from two residential communities of the same university nominate individuals with whom they were friends. A subset of the participants ($n = 63$) were then scanned using fMRI as they watched an assortment of video clips. The authors computed the intersubject correlation in brain activity during video-watching, and found that the neural responses of "central" individuals (i.e. individuals whom more participants had nominated as their friend) were more similar to that of the group. In addition, neural responses of high-centrality individuals were more similar to other high-centrality individuals, but neural responses of low-centrality individuals were not more similar to other low-centrality individuals. Together, these results suggest that individuals central in a social network respond to stimuli more similarly to their peers, but less central individuals respond more idiosyncratically.

The manuscript builds on growing work showing the neural similarity while watching dynamic audio/audio-visual stimuli captures similarity in how stimuli are interpreted, and tracks with other indexes of similarity (e.g., personality profiles and social distance). The current results make a novel contribution to our understanding of what differentiates individuals who are more likely to attract friends, and will be of broad interest to researchers who study social relationships and

social networks. The analyses are sensible and support the claims of the manuscript, and the appropriate control analyses (e.g., social distance, similarity in preferences/demographics) were performed. I do not have major concerns about the manuscript. Below are my comments that I hope the authors will find useful:

1. The authors performed a median split to identify high vs. low centrality participants. This median split, however, resulted in rather uneven groups (high = 23, low = 40). This raises some theoretical and methodological issues. First, this split is perhaps more accurately described as a tercile split rather than a median split, with the analyses effectively comparing the top third of participants against the bottom two thirds of participants. I think this should be made clear when describing the analyses and when discussing the results.
2. Relatedly, is there a linear relationship between centrality and mean neural similarity, or is there something special about being in the top third? I think it would be illustrative to show a scatterplot of mean neural similarity and centrality in some key ROIs to better visualize the relationship between the two variables.
3. Could the observation that low centrality participants have more idiosyncratic responses be driven by the unequal group sizes? In particular, a larger group is more likely to have diverse viewpoints (from having more people in the group), which could contribute to lower ISC within the group. I think the results would be more convincing if the authors repeated the analyses with equal-size groups.
4. The authors corrected for multiple comparisons by controlling for FDR. One issue with FDR correction is that it can be overly liberal (i.e., a critical p threshold at uncorrected $p > 0.05$) when there are a sufficient number of positive tests. To ensure that this is not the case, I think the authors should report the uncorrected p-value when reporting ROI results in the main text, and the corresponding critical p value (i.e. the uncorrected p-value at which adjusted $p < 0.05$) for each statistical map.

Response Letter

Manuscript: "Popularity is linked to neural coordination: Neural evidence for an Anna Karenina principle in social networks" [NCOMMS-21-21617]

Comments from Reviewer 1

1. *"The authors wanted to take a finer grain approach to test the Anna Karenina principle, and used dyad-level ISC analysis. I think this is a good approach to take. That said, I wonder why the authors divided the dyads into three groups (high-high, low-low and high-low), instead of using IS-RSA as suggested in one of the co-authors' paper (Finn, et al. NeuroImage, 2020)? I think it is a more sensitive and appropriate method to use for testing this."*

R1.1 Thank you for this comment. We made the decision to conduct our analyses using the three-group approach to parallel the approach of Finn et al. (2018), *Nature Communications*. We believe that the three-group analysis allows a more specific test of the Anna Karenina principle than the IS–RSA approach, as our chosen approach directly contrasts the {high, high} dyads to the {low, low} dyads (i.e., $ISC_{\{high, high\}} > ISC_{\{low, low\}}$), and $ISC_{\{high, high\}} > ISC_{\{low, low\}}$ should hold for results that reflect an Anna Karenina principle, but not for results that follow a nearest-neighbor model (i.e., the kind of model that is tested in most ISC studies, which reflects the assumption that individuals who are more similar in a behavioral trait have more similar neural responses).

On the contrary, results that support the hypothesis that {high, high} dyads have greater neural similarity than {low, high} dyads (i.e., $ISC_{\{high, high\}} > ISC_{\{low, high\}}$) would also be true for a nearest-neighbor model, and neural similarity does not necessarily have to be greater in {low, high} dyads than in {low, low} dyads (i.e., $ISC_{\{low, high\}} > ISC_{\{low, low\}}$) to support an Anna Karenina principle (e.g., if each individual in the low-centrality group responded in an entirely unique way, then they should have comparably low ISCs with other low-centrality individuals and with high-centrality individuals). Similarly, if using an IS–RSA approach, significant similarities between a model representational dissimilarity matrix (RDM) and a neural RDM could be driven largely by phenomena that are not specific to a pattern of results that follow an Anna Karenina principle. (For example, if there are relatively low similarities among dyads that consist of one high-centrality individual and one low-centrality individual.)

In summary, we believe that our method of contrasting the three different groups is a more direct test of the Anna Karenina principle than the IS–RSA approach, as $ISC_{\{high, high\}} > ISC_{\{low, low\}}$ holds only if the results follow an Anna Karenina principle. We have added the following sentences in the Results section (where we first discuss our approach) to make this point explicit:

"The $ISC_{\{high, high\}} > ISC_{\{low, low\}}$ contrast is our most direct test of the hypotheses that highly-central individuals have exceptionally similar neural responses to one another, whereas less-central individuals have neural responses that are idiosyncratic. This is the case because it tests whether neural similarity is greater in dyads in which both individuals had a high in-degree

centrality than in dyads in which both individuals had a low in-degree centrality. By contrast, $ISC_{\{high, high\}} > ISC_{\{low, high\}}$ would also hold for a nearest-neighbor model³¹, which reflects the assumption that individuals who are more similar in a behavioral trait also exhibit greater neural similarity, and $ISC_{\{low, high\}} > ISC_{\{low, low\}}$ does not necessarily have to hold to support the hypotheses that highly-central individuals have exceptionally similar neural responses to one another but that less-central individuals have neural responses that are idiosyncratic. For example, if each low-centrality subject responded in a completely unique way, then they would have similarly low ISCs with other low-centrality individuals and with high-centrality individuals). With that said, we reasoned that $ISC_{\{low, high\}} > ISC_{\{low, low\}}$ was likely to arise in the current dataset because of underlying stimulus-driven responses that are shared across all participants, and each low-centrality individual will partially reflect these shared stimulus-driven responses (even if they each deviate from the normative responses in an idiosyncratic way). Accordingly, we report the results of three contrasts: (1) $ISC_{\{high, high\}} > ISC_{\{low, low\}}$, which is the most direct test of our hypotheses; (2) $ISC_{\{high, high\}} > ISC_{\{high, low\}}$, which is a test of our hypotheses but also holds for a nearest-neighbor model; and (3) $ISC_{\{low, high\}} > ISC_{\{low, low\}}$, which does not have to hold to support our hypotheses, but which we expect to hold." (pg. 16-17)

We have also added a sentence to the caption of Fig. 4 (where we visualize our Anna Karenina results) to emphasize the $ISC_{\{high, high\}} > ISC_{\{low, low\}}$ contrast as the most direct test of our hypothesis:

"The $ISC_{\{high, high\}} > ISC_{\{low, low\}}$ contrast in (b) provides the most direct test of our main hypotheses that highly-central individuals have exceptionally similar neural responses to one another and that less-central individuals have neural responses that are idiosyncratic." (pg. 19)

Additionally, we also complement our main analyses (which use the binarized in-degree centrality variables) with a non-binarized version of in-degree centrality. To do so, we related the minimum in-degree centrality value for each dyad to neural similarity. This procedure implements the IS–RSA approach that was described in Finn et al. (2020), *NeuroImage*. In our initial submission, we had included these results in the Supplementary Information. Given your feedback, we have now moved these results from the Supplementary Information to the main manuscript (Fig. 5; pg. 20). For your convenience, we also include the relevant figure below.

Dyad-level results: Associating neural similarity with the minimum in-degree centrality in dyads

Fig. 5. Dyad-level associations of neural similarity with the minimum in-degree centrality of dyads. We found a positive association between ISC and minimum in-degree centrality. Larger ISCs in brain regions (including the DMPFC, the VLPFC, the precuneus, the temporal pole, and portions of the superior parietal lobule) were associated with a higher minimum in-degree centrality. The quantity B is the standardized regression coefficient. Regions where we observed significant associations between in-degree centrality and ISC are outlined in black. We used an FDR-corrected significance threshold of $p < 0.001$, which corresponds to an uncorrected p -value threshold of $p < 8.879 \times 10^{-5}$.

2. *“It seems that the distribution of the popular participants was not uniform, and most of them had 3-4 in-degree centrality. Was the similar-to-their peers effect stronger in the seven individuals that had 10 or more In-degree centrality?”*

R1.2 Thank you for the opportunity to clarify. Below we plot the associations between in-degree centrality and mean neural similarity with other participants in three key regions: right dorsomedial prefrontal cortex (rDMPFC), left dorsomedial prefrontal cortex (lDMPFC), and right precuneus (rPC). In these plots, the red dots represent the 7 individuals with in-degree centralities of at least 10. As you can see below, while neural similarity in these key regions increases with in-degree centrality, it does not seem to be the case that the individuals with in-degree centralities of 10 or greater have dramatically higher neural similarity with their peers than other individuals with greater-than-median in-degree centrality. Although we hesitate to make strong claims due to the small sample size at the upper end of the distribution of in-degree centrality, one possibility is that the features that distinguish individuals with extremely high in-degree centrality may include other trait-level variables beyond normative processing. (For example, they may have more free time to make more friends, a greater need to belong,

and so on.) In other words, although processing the world in a relatively normative way may play a role in determining whether individuals will form enough social ties to be integrated into a community, other factors may also be important for determining if one regularly socializes with an exceptionally large number of people.

Note: We added jitter to the scatter plot for visualization purposes.

3. “It was not clear to me from the analysis description – were there regions in which the low centrality individuals were more synchronized than the high centrality individuals?”

R1.3 Thank you for the opportunity to clarify this point. Across all analyses, there were no brain regions where individuals with low in-degree centrality were more synchronized than individuals with high in-degree centrality. One can see these results in figures such as Fig. 4b, which we include below for convenience. If individuals with low in-degree centrality were more synchronized than individuals with high in-degree centrality in a given brain region, then the region would be in a shade of blue and also outlined in black if it was statistically significant. As our figures (such as Fig. 4b) indicate, however, no region was identified to be more similar in individuals with low in-degree centrality (as shown by the absence of blue-shaded regions that are outlined in black).

For clarity, we have added the following sentences in the Results section, where we discuss the results from our subject-level and dyad-level analysis.

In the discussion of our subject-level analysis: *“Additionally, in these analyses and in all of our other analyses, we did not find any regions in the brain in which low in-degree centrality was associated with more-similar neural responses to one’s peers.”* (pg. 11)

In the discussion of our dyad-level analysis: *“Across all of our analyses, we did not find any regions in the brain in which there were larger ISCs in {low, low} dyads than in {high, high} dyads. We also did not find any regions in the brain in which there were larger ISCs in {low, high} dyads than in {high, high} dyads, nor any in which there were larger ISCs in {low, low} dyads than in {low, high} dyads.”* (pg. 17-18)

4. *“If I understand Figure 3 correctly, it seems that the effect was much more pronounced in Community 2 than Community 1. Is this the case? And if so, do the authors have an explanation for this?”*

R1.4 Thank you for the opportunity to clarify this point. We conducted additional analyses to test if any of the effects that we observed between in-degree centrality and mean neural similarity in the rDMPFC, IDMPFC, and the rPC were more pronounced in one community or the other. To do so, we fit one additional generalized linear model (GLM) for each brain region with the ISC in the respective brain region as the dependent variable and three independent variables: the binarized in-degree centrality, the community number, and the interaction between the binarized in-degree centrality and the community number. None of the interaction terms were significant in any of the models (even without correcting for multiple comparisons across brain regions), indicating that the associations between ISC in these three regions and in-degree centrality were not more pronounced for one community or the other.

We would be happy to integrate any of these results into the Supplementary Information if you think it would be helpful.

Dependent variable: neural similarity in the rDMPFC

Variable	B	SE	p (uncorrected for multiple comparisons)
Intercept	-0.139	0.202	0.491
In-degree centrality	1.020	0.355	0.006**
Community × In-degree centrality	-0.013	0.466	0.977
Community	-0.415	0.278	0.140

Reference levels: high in-degree centrality; community 2. ** $p < 0.01$

Dependent variable: neural similarity in the IDMPFC

Variable	B	SE	p (uncorrected for multiple comparisons)
Intercept	-0.195	0.205	0.344
In-degree centrality	0.997	0.361	0.008**
Community × In-degree centrality	-0.013	0.475	0.978
Community	-0.298	0.283	0.296

Reference levels: high in-degree centrality; community 2. ** $p < 0.01$

Dependent variable: neural similarity in the rPC

Variable	B	SE	p (uncorrected for multiple comparisons)
Intercept	-0.021	0.210	0.922
In-degree centrality	0.737	0.370	0.051
Community × In-degree centrality	0.180	0.486	0.713
Community	-0.519	0.289	0.078

Reference levels: high in-degree centrality; community 2

We also fit additional analogous GLMs using the non-binarized in-degree centrality variables instead of the binarized in-degree centrality variables. Similar to our results with the binarized in-degree centrality variables, we found that none of the interaction terms were significant (even without correcting for multiple comparisons across brain regions).

Dependent variable: neural similarity in the rDMPFC

Variable	B	SE	p (uncorrected for multiple comparisons)
Intercept	0.232	0.181	0.205
In-degree centrality	0.477	0.197	0.018*
Community × In-degree centrality	-0.306	0.250	0.226

Community	-0.396	0.243	0.108
-----------	--------	-------	-------

Reference level: community 2. * $p < 0.05$

Dependent variable: neural similarity in the IDMPFC

Variable	B	SE	p (uncorrected for multiple comparisons)
Intercept	0.162	0.186	0.388
In-degree centrality	0.404	0.202	0.050
Community \times In-degree centrality	-0.266	0.257	0.305
Community	-0.272	0.249	0.280

Reference level: community 2

Dependent variable: neural similarity in the rPC

Variable	B	SE	p (uncorrected for multiple comparisons)
Intercept	0.244	0.186	0.195
In-degree centrality	0.309	0.202	0.133
Community \times In-degree centrality	-0.181	0.257	0.485
Community	-0.427	0.250	0.093

Reference levels: community 2

5. *“The authors performed an important analysis that found that individuals’ enjoyment and interest from the clips did not explain the popularity effect on the neural synchronization. However, it could still be that similar interpretation of the clips (which is not reflected in enjoyment and interest) was what drove the increased neural synchronization of popular individuals. How can the authors rule out this possibility?”*

R1.5 Thank you for this comment. We agree with you that similarities in interpretation of the clips that are not captured by our enjoyment and interest ratings could be at least partially driving neural similarity in popular individuals. One advantage of neuroimaging is that we are able to capture similarities in many different types of processing simultaneously that may be challenging to obtain through self-reports for various reasons. For instance, asking participants to rate each stimulus on a large number of different dimensions could increase participant

burden and require more effortful thinking that can disrupt natural processing. It could also be difficult to identify the types of dimensions that are likely to be important in the effects of interest. To that end, using neuroimaging can help generate hypotheses about the types of self-report measures that may be sensitive to the kinds of interpersonal similarities that may be associated with a trait such as popularity. Findings from our study suggest that neural similarities in regions of the default-mode network are associated with popularity. Given that these regions have been associated with social cognitive functions and high-level interpretation of events, future work that tests the associations between similarities in self-report ratings that capture individuals' social and cognitive tendencies and/or free-response measures that capture nuances in individuals' processing of stimuli (e.g., semantic analyses of open-ended descriptions) with popularity may be particularly fruitful.

We discuss these ideas in the following paragraph in the Discussion section. We have also added a few sentences (which we show in bold below) at the end of the paragraph to explicitly address the possibility that other types of self-report measures could capture the types of similarities in interpretation that may be associated with popularity:

*“Notably, controlling for similarities in the enjoyment and interest ratings did not change our results that link neural similarity with popularity. That is, we found that neural similarity in brain regions that have been implicated in high-level interpretation and social cognition was associated with network centrality above and beyond what we were able to capture using self-reported preferences. This suggests that measuring neural responses to naturalistic stimuli as they unfold over time can capture consequential aspects of mental processing beyond what one can obtain using a few targeted self-report questions. The strong link between popularity and ISCs (even when controlling for similarities in participants' self-reported preferences), relative to links between similarities in popularity and self-reported preferences, may arise from several factors, including the finer temporal granularity of ISCs than our self-report measures (because ISCs capture similarities in how responses evolve over time), the limits of self-report (because people are often unaware of and/or unwilling to report features of their attitudes and other aspects of their mental processing⁴⁷), and the possibility that the similarities in processing that are linked to centrality reflect similarities in the creation of internal models of situations as they evolve over time (rather than reflecting similarities in what participants found interesting or enjoyable)⁴². A notable benefit calculating ISCs is that one can use them to characterize similarities in many different aspects of mental processing in parallel, and one can thereby obtain insight into diverse emotional and cognitive processes that unfold in response to various situations and which may be shaped by individuals' pre-existing beliefs, values, attitudes, and experiences. **Our neural findings also provide insights that can inform which self-report measures are likely to capture the types of processing that may be particularly similar among popular individuals. It may be particularly fruitful to test for associations between popularity and the normativity of (1) self-report scales that capture individuals' social and cognitive tendencies and/or (2) individuals' processing of stimuli (e.g., as captured by semantic analyses of free-response measures).**” (pg. 24-26)*

6. *“The authors used 14 video clips, which they chose in order to elicit meaningful variability in the interpretations and meaning that different individuals can draw from the content. Were there specific contents that more powerfully generated the “popularity neural synchronization effect?””*

R1.6 Thank you for this question. In the present document, we include two tables that show subject-level and dyad-level results that relate ISCs in the rDMPFC parcel that was robustly implicated across analyses (see, e.g., Supplementary Figures 3, 4, and 6) with the binarized in-degree centrality variable for each video. Given that we designed our study to detect subject-level effects, rather than video-level effects, we urge caution when interpreting these exploratory results. For one thing, all participants saw the video clips in the same order in an effort to avoid any potential variability in neural responses from differences in the way that the stimuli were presented (rather than from endogenous participant-level differences). Therefore, any stimuli-level effects are confounded by the order in which stimuli were presented. For example, it is possible that a video would have evoked higher or lower ISCs because it was presented before and after specific videos and/or at a point in the study when participants tended to be more attentive. Additionally, the duration of stimuli may also be a potential confounding factor, given that neural responses in higher-level cognition regions (such as regions of the default-mode network, where our effects were most pronounced) track information that unfolds over longer time frames. (See prior work on temporal receptive windows; Lerner et al., 2011.) Notably, the tables below show that the association between ISCs in the rDMPFC with in-degree centrality is the largest in our two longest clips. (These are videos 11 and 12; see Supplementary Table 1 for further descriptions of the videos.) These two video clips are also distinct from the other clips that we used in the study in that they feature ambiguous scenes and themes (e.g., those that have more room for different interpretations and meaning for different audiences) that may allow deeper processing of social meaning related to ISCs that are involved in high-level processing. Accordingly, it is possible that stimuli that feature ambiguous social-narrative arcs may accentuate subject-level differences in higher-level processing that may distinguish popular individuals from less-popular individuals. Future work that is designed to confer sensitivity to stimuli-level effects can provide further insights into the processes at play.

Given the limitations that we note above, we have tentatively decided to not include the tables below from the manuscript itself. However, we would be happy to integrate any of these results into the Supplementary Information if you think it would be helpful.

Subject-level results that relate ISCs with the binarized in-degree centrality (high versus low):
Video-level results for rDMPFC

Video	B	SE	p	Duration (MM:SS)
Video 1	0.514	0.255	0.188	03:43
Video 2	0.441	0.258	0.258	05:05
Video 3	0.022	0.264	0.933	01:59
Video 4	0.503	0.256	0.188	02:58
Video 5	0.353	0.260	0.313	03:22
Video 6	-0.359	0.260	0.313	03:15
Video 7	0.403	0.259	0.290	01:31

Video 8	0.324	0.261	0.339	02:49
Video 9	-0.079	0.264	0.339	01:46
Video 10	0.037	0.264	0.933	01:41
Video 11	0.641	0.252	0.095†	08:28
Video 12	0.737	0.247	0.059†	12:14
Video 13	0.236	0.263	0.523	03:30
Video 14	0.026	0.265	0.933	07:16

† $p < 0.10$; we have FDR-corrected all p -values because of multiple comparisons

Dyad-level results that relate ISC with binarized in-degree centrality: Video-level results for rDMPFC
 Contrast: $ISC_{\{high, high\}} > ISC_{\{low, low\}}$

Video	B	SE	p	Duration (MM:SS)
Video 1	0.222	0.104	0.013*	03:43
Video 2	0.210	0.138	0.089†	05:05
Video 3	0.022	0.161	0.888	01:59
Video 4	0.362	0.162	0.010*	02:58
Video 5	0.209	0.172	0.162	03:22
Video 6	-0.197	0.150	0.127	03:15
Video 7	0.203	0.125	0.070†	01:31
Video 8	0.247	0.187	0.127	02:49
Video 9	-0.073	0.128	0.549	01:46
Video 10	0.068	0.160	0.680	01:41
Video 11	0.484	0.185	0.002**	08:28
Video 12	0.643	0.208	< 0.001***	12:14
Video 13	0.176	0.155	0.167	03:30
Video 14	0.002	0.175	0.986	07:16

† $p < 0.10$, * $p < 0.05$, ** $p < 0.01$; we have FDR-corrected all p -values because of multiple comparisons

7. “Did the authors test if popular individuals were more similar in their brain responses to the individuals that nominated them than to other individuals? In a way, this could be used as a replication to Parkinson et al 2018 paper on “similar neural responses predict friendship”.

R1.7 Thank you for this question. In the current paper, we unfortunately do not have the power to test whether popular individuals have more similar neural responses to those that nominated them, given that only a subset of the full network (specifically, the people who completed the social-network survey) participated in the fMRI study. Therefore, there were relatively few directly connected dyads (i.e., friends) in the fMRI sample and even fewer directly-connected dyads in subsets of the sample (e.g., divided according to popularity). We have conducted separate analyses that test whether individuals who are closer (which we determined based on a median split of social distances between dyads in the fMRI sample) in the social network have more similar neural responses than individuals who are further apart in the social network. Replicating Parkinson et al., (2018), we found that people who were closer in the social network had higher ISCs in regions (such as the inferior parietal lobules, precuneus, and the medial prefrontal cortex), that are associated with high-level cognition. See below for a figure of these results. We are not planning to include these results, as they are beyond the scope of the present paper, and these results are currently being prepared for a different manuscript in progress. That said, these results are consistent with the idea that individuals who are closer in a social network have more-similar neural responses than individuals who are further apart in a social network.

8. “Did the authors measured participants’ conformity? It could be interesting to relate one’s conformity to being similar to the group’s norm neural response.”

R1.8 Thank you for this suggestion. We agree that it would be interesting to relate individual differences in conformity to similarity to a group’s normative neural response. In the current study, we unfortunately did not have a measure of conformity. We have added a sentence (which we show in bold below) in the Discussion section that directly discusses the potential value of a future study that investigates the relationship between conformity and neural similarity in popular individuals.

*“Because the present study has only one wave of fMRI data, we are not able to ascertain the causal mechanisms that drive our effects. Prior research suggests that popular individuals have more behavioral and neural sensitivity than unpopular people to interpersonal cues⁴⁸ and that highly-central individuals are more likely than less-central individuals to adapt their brain activity to match that of other individuals in their social group⁴⁹. Therefore, one possibility is that people who become popular may adapt their views of the world to meet their social network’s normative ways of processing the world, perhaps due to a greater need to belong socially or a desire to connect with a large number of people. Future studies that employ longitudinal data can help elucidate the direction(s) of these effects and further clarify the mechanisms that may be at play. **For instance, pairing in-lab measures of social conformity with longitudinal studies that examine neural similarity may provide insight into the extent to which individual differences in in-lab measures of conformity are predictive of changes in peoples’ unconstrained processing of the world around them.**” (pg. 27)*

Comments from Reviewer 2

1. “The main question being investigated here is whether people in central positions within social networks—i.e., those who are well-connected to many others in the network—process naturalistic stimuli in was similar to their peers as compared to people who are less-central in social networks. It is stated that since ‘personality traits’ (i.e., extraversion, emotional stability) are inconsistently associated with being well-connected, the role of centrality within social networks may be more important to understanding and connecting with others. Yet, could there be individual differences in other characteristics that could serve as the underlying mechanism here driving the similarity in neural processing of naturalistic stimuli. For example, is it just that they are central in the network? Or is it that they all have similar levels of social abilities/function. Do the authors have any additional data on anything like mentalizing, empathy, interpersonal emotion regulation or the like that might help tease apart or further explain what is driving this network centrality effect? Again, I really like this study, but there just feels like there is some piece that is missing here.”

R2.1 Thank you for bringing up this important point. We agree that individual differences in social abilities or functioning (or other traits or tendencies) may be important mechanisms that link social-network centrality and neural similarity. Unfortunately, in the current paper, we did not

gather additional data to obtain a measure of individual differences in social functioning. We do agree that these are important considerations to paint a more nuanced picture of relationships between centrality and neural similarity and that future work can further elucidate the mechanisms at play.

We added a paragraph in the Discussion to explicitly address these limitations of the current study and suggest that these are important considerations that future studies can explore.

“Another possibility is that popular individuals may show similarly high levels of social abilities and functioning, which may in turn lead to greater neural similarity. For instance, it is possible that popular individuals may have distinctively high levels of empathic concern, mentalizing abilities, and/or emotion-regulation abilities that help them form and maintain a large number of social ties and also impact how they respond to naturalistic stimuli. Therefore, potential differences in social functioning between popular and other individuals may be a key reason for the links between popularity and neural similarity that we found in the present study. We did not collect measures that can capture individual differences in social functioning, so we are not able to test these theories using our data. Future studies that investigate associations between individual differences in social functioning, centrality, and neural processing of naturalistic stimuli can further elucidate these relationships.” (pg. 27-28)

2. *“The authors collected ratings of enjoyment and interest in the movie clips that participants viewed, and report similar patterns of results in terms of network centrality and subjective perceptions of the videos. They also used these ratings in a GLM to see whether the effect of network centrality held when accounting for the variance contributed by these ratings. However, I think it might also be useful to know whether these ratings indeed predicted neural similarity in the same regions (i.e., DMN) or others? Or not at all?”*

R2.2 Thank you for this question. We did find that similarities in interest and enjoyment ratings were associated with neural similarity in subregions of the default-mode network (DMN) when not controlling for in-degree centrality. (See the figure below.)

Associations between similarities in preference ratings with neural similarity, no control variables

However, when we controlled for dyad-level binarized in-degree centrality by including the variable as a covariate, the associations between similarities in our preferences ratings and ISCs in many of the parcels were no longer significant. (See the figure below.)

Notably, our binarized in-degree centrality results remain robust even after controlling for similarities in self-reported preferences. (See Supplementary Figure 13, which we also include below for comparison.) Furthermore, the pattern of results is more consistent and widespread, as indicated by warm colors throughout the brain. In other words, although the relationship between ISCs and social-network centrality remains relatively consistent and robust after accounting for these self-report ratings, the relationship between ISCs and similarities in self-report ratings becomes less consistent and robust after accounting for social-network centrality.

We have included additional figures in the Supplementary Information (specifically, Supplementary Figs. 11 and 12) that illustrate the associations between similarities in self-reported preferences and neural similarity. We have also noted this in the main manuscript (which we show in bold):

“Although similarity in enjoyment and interest ratings were also associated with neural similarity in regions of the default-mode network (see Supplementary Figs. 10 and 11), our results indicate that the “Anna Karenina” pattern of results that links ISCs and dyad-level in-degree centralities remains significant after controlling for similarity in enjoyment and interest ratings (see Supplementary Fig. 13). Therefore, we conclude that our findings that greater neural similarity tends to occur between highly-central individuals and that reduced neural similarity tends to occur between less-central individuals arise from differences beyond those that were captured by self-reported preference ratings.” (pg. 21-22)

We found similar results when we used our non-binarized in-degree centrality variable instead of the binarized variable. Analogous to what we found when we controlled for binarized in-degree centrality in relating similarities in preference ratings and ISCs, when we controlled for the non-binarized in-degree centrality variable, the associations between similarities in our preferences ratings and ISCs in many of the parcels were no longer significant. Notably, our minimum in-degree centrality results remain robust even after controlling for similarities in self-reported preferences. Furthermore, the patterns in the results are more consistent and widespread (as indicated by the warm colors throughout the brain), and the effect sizes are larger in our results that associate centrality with ISCs than in our results that associate similarities in preference ratings and ISCs. (See the figure below.)

3. “Relatedly, I was just curious about the decision to hold all ratings of enjoyment and interest in the movie clips until the end of the task, as opposed to querying participants right after the presentation of each stimulus?”

R2.3 Thank you for the opportunity to clarify this point. We obtained these ratings at the end of the task rather than during the task in an effort to minimize potential biases in processing that may occur if participants were asked to reflect on the content immediately after each stimulus. We have added our rationale for this decision to the Methods section:

“We obtained these preference ratings after the fMRI scan in an effort to minimize potential biases or disruptions in processing that could occur if participants were asked to reflect on content immediately after each stimulus was presented during scanning.” (pg. 33)

4. “The authors report a final N of 63 for the fMRI component of their study after exclusions, initially starting at 70. However, in reading on p28 (starting on line 566) details regarding subject exclusions, the numbers do not seem to easily add up to a total of 7 participants excluded. Can the authors clarify?”

R2.4 Thank you for this opportunity to clarify this point. We have revised the wording (which we show in bold) where we discuss the exclusions and hope that it is clearer:

*“A total of 70 participants from the aforementioned two residential communities participated in the neuroimaging portion of our study. **We excluded two individuals due to excessive movement in more than half of the scan and excluded one individual who fell asleep during half of the scan. We also excluded one individual who did not complete either the scan or the social-network survey. Three additional fMRI participants did not complete the social-network survey. (See Supplementary Table 9 for a table of excluded participants.)** This resulted in a total of 63 participants (40 female) between the ages of 18 and 21 (with a mean age of $M = 18.19$ and a standard deviation of $SD = 0.59$) that we included for all analyses.”* (pg. 31)

For further clarity, here is a table of excluded participants. We have also included this table in the Supplementary Materials (Supplementary Table 9).

Table of excluded participants

Participant	Reason(s)
Excluded participant 1	Excessive movement during the fMRI scan
Excluded participant 2	Excessive movement during the fMRI scan
Excluded participant 3	Fell asleep during half of the fMRI scan
Excluded participant 4	Did not complete the fMRI scan and did not complete the social-network survey
Excluded participant 5	Did not complete the social-network survey
Excluded participant 6	Did not complete the social-network survey
Excluded participant 7	Did not complete the social-network survey

5. *“In the section on ‘Subject-level preference analysis’ (p13-14), the authors note that people who were more popular were more similar than less popular people in what they found to be enjoyable and interesting; the result for what they found ‘interesting’ is trend-level, and this should be acknowledged.”*

R2.5 Thank you for pointing this out. We have included a sentence (which we show in bold) where we discuss our subject-level preference results:

*“Our results indicate that individuals who were more popular in their social networks were more similar, on average, than less-popular individuals with their peers in the content that they found to be enjoyable ($B = 0.578$, $SE = 0.253$, $p = 0.026$) and interesting ($B = 0.491$, $SE = 0.256$, $p = 0.061$). **Note that the association between popularity and mean interest similarity is only marginally statistically significant (i.e., trend-level.)**”* (pg. 14)

6. *“The authors presented the video clips in the same order for all participants to ‘avoid any potential variability in neural responses from differences in the way that the stimuli were presented’. Yet, I think the argument could be made that without multiple orders, it is difficult to know whether the results are due solely to the way in which the stimuli appeared for everyone. I would think this should also be acknowledged as a potential limitation, even though it was intended to be an advantageous design (and it may well be).”*

R2.6 Thank you for this comment. Our consistently-ordered series of stimuli can be thought of as a single continuous stream of content, analogous to different scenes in a movie (although the content was less related across clips in our study than across scenes in a typical movie). We agree that different relative orderings of the stimuli could generate different results, similar to how reordering scenes in a movie might generate different results in that differences in how tone, narratives, and themes evolve over a time period can lead to differences in high-level processing. It is possible that one may obtain stronger or weaker effects if the clips are assembled differently. As you have noted, we presented the video clips in the same order for all participants because our main priority was ensuring the same overall context of viewing the video clips for everyone within the study; this choice allowed us to maximize sensitivity to subject-level differences. We clarified our rationale further by adding a few sentences in the Methods section (which we show in bold), where we discuss our decision to present the videos in the same order for all participants:

*“All participants saw the videos in the same order to avoid any potential variability in neural responses from differences in the way that the stimuli were presented (rather than from endogenous individual-level differences). **One can think of our consistently-ordered series of stimuli as a single continuous stream of content (analogous to different scenes in a movie). It is possible that different relative orderings of the stimuli could generate different results, similar to how reordering scenes in a movie might generate different results (e.g., due to differences in how tone, narratives, and themes evolve over the span of the movie). In our study, we presented the videos in the same order to all participants in order to keep the context surrounding each video consistent across participants because our main priority was to maximize sensitivity to individual-level differences.**”* (pg. 32-33)

Comments from Reviewer 3

1. *“The authors performed a median split to identify high vs. low centrality participants. This median split, however, resulted in rather uneven groups (high = 23, low = 40). This raises some theoretical and methodological issues. First, this split is perhaps more accurately described as a tercile split rather than a median split, with the analyses effectively comparing the top third of participants against the bottom two thirds of participants. I think this should be made clear when describing the analyses and when discussing the results.”*

R3.1 Thank you for this feedback. We have added a few sentences (which we show in bold) in the Results section when we first discuss the median split to explicitly address this issue. To mitigate these concerns, we have also moved the dyad-level results that relate the non-binarized version of in-degree centrality with neural similarity, which complement our median-split results, to the main manuscript from the Supplementary Information (see Fig. 5, pg. 20).

We also include this figure below for your convenience.

*“In our fMRI study, we classified participants as part of the high-centrality group if they had an in-degree that was larger than the median (specifically, if it was more than 2; there were $n_{high} = 23$ such people) and into the low-centrality group if they had an in-degree that was less than or equal to the median (specifically, if it was less than or equal to 2; there were $n_{low} = 40$ such people). See Supplementary Fig. 1 for plots of the in-degree distributions. **Because the median-split approach resulted in unevenly-sized groups, we also conducted additional analyses** to examine the relationships between the original non-binarized version of centrality and neural similarity whenever possible, as we describe in more detail below. **We also conducted analogous exploratory analyses with approximately equal-sized groups by contrasting individuals with in-degree centralities in the top and bottom thirds of the distribution. This yielded similar results to our main findings. See “Subject-level ISC analysis” and “Dyad-level ISC analysis” for more details.**” (pg. 7-8)*

Fig. 5. Dyad-level associations of neural similarity with the minimum in-degree centrality of dyads. We found a positive association between ISC and minimum in-degree centrality. Larger ISCs in brain regions (including the DMPFC, the VLPFC, the precuneus, the temporal pole, and portions of the superior parietal lobule) were associated with a higher minimum in-degree centrality. The quantity B is the standardized regression coefficient. Regions where we observed significant associations between in-degree centrality and ISC are outlined in black. We used an FDR-corrected significance threshold of $p < 0.001$, which corresponds to an uncorrected p -value threshold of $p < 8.879 \times 10^{-5}$.

Please also see **R3.3** on pages 23–24 of this document for results from analogous models to our main analyses (involving the binarized in-degree centrality variable) that use approximately equal-sized groups.

2. *“Relatedly, is there a linear relationship between centrality and mean neural similarity, or is there something special about being in the top third? I think it would be illustrative to show a scatterplot of mean neural similarity and centrality in some key ROIs to better visualize the relationship between the two variables.”*

R3.2 Thank you for this question. We include plots below that visualize the relationships between centrality and mean neural similarity in three key brain regions (rDMPFC, IDMPFC, and rPCC). As you can see in these plots, although neural similarity in these key regions is larger for individuals with larger in-degree centralities, it does not seem to be the case that the individuals who are in the top third of in-degree centrality have exceptionally similar neural responses to their peers (relative to other individuals whose in-degree centralities are larger than the median). Although we hesitate to make strong claims because of the small sample size at the upper end of the distribution of in-degree centrality, one possibility is that the features that distinguish individuals with extremely high in-degree centralities may include other trait-level variables beyond normative processing. (For example, they may have more free time to make more friends, they may have a higher need to belong, and so on.) In other words, although processing the world in a relatively normative way may play a role in determining whether an individual forms enough social ties to be integrated into a community, other factors may also be important for determining if they regularly socialize with an exceptionally large number of people.

Note: We added jitter to the scatter plot for visualization purposes.

We would be happy to include these plots in the Supplementary Information if you think that it would be helpful.

3. *“Could the observation that low centrality participants have more idiosyncratic responses be driven by the unequal group sizes? In particular, a larger group is more likely to have diverse viewpoints (from having more people in the group), which could contribute to lower ISC within the group. I think the results would be more convincing if the authors repeated the analyses with equal-size groups.”*

R3.3 Thank you for this question. As we mentioned in response **R3.1** on pg. 20-21 of this document, in order to address similar concerns, we report results from analyses that are analogous to our main analyses, except that they use non-binarized versions of the in-degree centrality variable for both the subject-level and dyad-level analyses (see Fig. 2d and Fig. 5). These results replicate our results using the binarized versions of in-degree centrality.

That said, please also see our subject-level and dyad-level results below that use the binarized in-degree centrality variable with approximately equal-sized groups. As you noted, using the median split in the full data leads to unequal group sizes, with 23 people in the “high” group and 40 people in the “low” group. To address your question, we repeated our analyses using a subset of our data by removing individuals with the median in-degree centrality value, which led to 23 people in the “high” group and 26 people in the “low” group. As you can see below, the results of these analyses not only replicate our main findings but actually have larger effect sizes that link ISCs with centrality.

We have incorporated these figures into the Supplementary Information (see Supplementary Figs. 5 and 10). We have also incorporated these findings into the main manuscript in the following places (which we show in bold) in the “Results” section:

*“Because the median-split approach resulted in unevenly-sized groups, we also conducted additional analyses to examine the relationships between the original non-binarized version of centrality and neural similarity whenever possible, as we describe in more detail below. **We also conducted analogous exploratory analyses with approximately equal-sized groups by contrasting individuals with in-degree centralities in the top and bottom thirds of the distribution. This yielded similar results to our main findings. See “Subject-level ISC analysis” and “Dyad-level ISC analysis” for more details.**”* (pg. 8)

*“We also fit analogous models to control for demographic variables that may be associated with neural similarity^{24,39}, models that only incorporated neural similarities between subjects who were living in the same residential community, models that controlled for social distances between participants in the same community, and **models that used a subset of the data with approximately equal-sized centrality groups.** These other approaches yielded similar results. See Supplementary Figs. 2–5.)”* (pg. 10)

*“**We also report the results of models that use a subset of the data with approximately equal-sized centrality groups (see Supplementary Fig. 10).**”* (pg. 17)

Supplementary Fig. 5. Subject-level results with approximately equal-sized in-degree centrality groups. The associations between ISCs in the right and left DMPFC and in-degree centrality remain significant when we used a subset of the data with approximately equal-sized in-degree centrality groups. Additionally, we found that the ISCs in other regions (including the precuneus and the inferior parietal lobule) of the default-mode network were also associated with in-degree centrality. The quantity B is the standardized regression coefficient. Regions where we observed significant associations between in-degree centrality and ISC are outlined in black. We used an FDR-corrected significance threshold of $p < 0.05$, which corresponds to an uncorrected p -value threshold of $p < 0.002$.

Supplementary Fig. 10 for “Dyad-level ISC analysis” in the “Results” section

Supplementary Fig. 10. Dyad-level associations of neural similarity with in-degree centrality with approximately equal-sized in-degree centrality groups. We identified similar brain regions as significantly associated with in-degree centrality as in our results in the main manuscript (see Fig. 4) when we used a

subset of the data with approximately equal-sized centrality groups. The quantity B is the standardized regression coefficient. Regions where we observed significant associations between in-degree centrality and ISC are outlined in black. We used an FDR-corrected significance threshold of $p < 0.001$, which corresponds to an uncorrected p-value threshold of $p < 0.0001$.

4. *“The authors corrected for multiple comparisons by controlling for FDR. One issue with FDR correction is that it can be overly liberal (i.e., a critical p threshold at uncorrected $p > 0.05$) when there are a sufficient number of positive tests. To ensure that this is not the case, I think the authors should report the uncorrected p-value when reporting ROI results in the main text, and the corresponding critical p value (i.e. the uncorrected p-value at which adjusted $p < 0.05$) for each statistical map.”*

Thank you for the opportunity to clarify this issue. All of our FDR-corrected p-values are smaller than the uncorrected p-values. Per your suggestion, we have added the corresponding uncorrected p-values alongside the corrected p-values when we report ROI results in the main text, and we have added the corresponding critical p-value to each of our statistical maps. We have also added a third column with corresponding uncorrected p-values in the Supplementary Tables where we report subcortical results.

For example, we have added the corresponding uncorrected p-values (which we show in bold) where we report our subject-level analyses:

*“We found that neural similarities in the bilateral DMPFC (left DMPFC: $\rho = 0.420$, $p_{corrected} = 0.048$, **$p_{uncorrected} < 0.001$** ; right DMPFC: $\rho = 0.415$, $p_{corrected} = 0.048$, **$p_{uncorrected} < 0.001$**), precuneus ($\rho = 0.408$, $p_{corrected} = 0.048$, **$p_{uncorrected} < 0.001$**), and left superior parietal lobule ($\rho = 0.424$, $p_{corrected} = 0.048$, **$p_{uncorrected} = 0.002$**) were significantly correlated with in-degree centrality (see Fig. 2d).”* (pg. 11)

Additionally, we have added the corresponding critical p-value to each of our statistical maps in the figure captions, such as in Fig. 2 (on pg. 12; copied below for your convenience):

Fig. 2. Subject-level analysis. (a) Our approach for subject-level analysis. First, we Fisher z-transformed the dyad-level inter-subject correlations, which are encoded by a matrix of pairwise Pearson correlation coefficients (which we denote by r). We then computed the mean of each subject's ISC with each other subject. (In other words, we took the mean of each row of the matrix.) We performed the above calculations for each of the 214 brain regions. This yields one ISC value for each subject for each brain region. The ISC value encodes the mean similarity in neural responses between the subject and each other subject in the corresponding brain region. (b) We tested for relationships between the subjects' in-degree centrality and these subject-level ISC values in each brain region. (c) Our results that relate mean ISCs with the binarized in-degree centrality variable indicated that individuals with high in-degree centrality had a much larger mean neural similarity with their peers in the bilateral DMPFC and precuneus than individuals with a low in-degree centrality. (d) Our results that relate mean ISCs with the original (i.e., non-binarized) in-degree centrality values gave similar results as the analysis in (c). We found that the mean ISCs in the bilateral DMPFC, precuneus, and the superior parietal lobule were positively correlated with in-degree centrality. The quantity B denotes the standardized regression coefficient, and ρ denotes the Spearman rank

correlation. All results are FDR-corrected at $p < 0.05$, which corresponds to an uncorrected p -value of 0.009 in (c) and an uncorrected p -value of 0.001 in (d).

Supplementary Table 2. Subject-level results that relate ISCs with the binarized in-degree centrality (high versus low): Subcortical results

Subcortical region	B	SE	p (corrected)	p (uncorrected)
Accumbens (L)	0.363	0.260	0.325	0.167
Amygdala (L)	0.578	0.253	0.164	0.026
Caudate (L)	0.269	0.262	0.466	0.307
Hippocampus (L)	0.550	0.254	0.180	0.031
Pallidum (L)	0.611	0.252	0.144	0.018
Putamen (L)	0.300	0.261	0.415	0.256
Thalamus (L)	0.432	0.258	0.247	0.099
Accumbens (R)	0.080	0.264	0.834	0.764
Amygdala (R)	0.493	0.256	0.210	0.059
Caudate (R)	0.171	0.263	0.640	0.519
Hippocampus (R)	0.587	0.253	0.158	0.024
Pallidum (R)	0.068	0.264	0.845	0.798
Putamen (R)	0.302	0.261	0.414	0.252
Thalamus (R)	0.482	0.256	0.220	0.065

We have FDR-corrected all p -values because of multiple comparisons; we also report the corresponding uncorrected p -values. The quantity B is the standardized regression coefficient, and the quantity SE is the standard error.

REVIEWERS' COMMENTS

Reviewer #1 (Remarks to the Author):

The authors have adequately addressed my concerns. I find the analyses clearer and believe the manuscript merits publication in Nature Communications.

Reviewer #2 (Remarks to the Author):

The authors have more than adequately addressed all of my comments regarding their initial submission in the revised manuscript. I believe the manuscript is clearer and stronger in its current, revised form. I have no further concerns.

Reviewer #3 (Remarks to the Author):

The authors have addressed my concerns and I am happy to recommend publication. I believe this paper will be impactful, and be of broad interest to researchers who study social relationships and social networks.

I do have a quibble about R3.2 in the response letter. I don't think one can examine non-linearity by plotting the ranks of the two variables. A monotonic non-linear relationship will appear as linear when transformed into rank orders. While I don't think it will affect the main claims of the manuscript, I do think it will be informative to examine if the relationship is linear or non-linear, and would encourage the authors to consider adding a scatter plot of the variables (either in their original units, or in z-scores) to the supplementary material.

Response Letter

Manuscript: "Popular individuals process the world in particularly normative ways" [NCOMMS-21-21617A]

Comments from Reviewer 1

"The authors have adequately addressed my concerns. I find the analyses clearer and believe the manuscript merits publication in Nature Communications."

Thank you so much for your helpful comments and feedback. We are delighted that you found the analyses to be clearer and the manuscript to be suitable for publication.

Comments from Reviewer 2

"The authors have more than adequately addressed all of my comments regarding their initial submission in the revised manuscript. I believe the manuscript is clearer and stronger in its current, revised form. I have no further concerns."

Thank you so much for your helpful feedback and insight. We are glad that we were able to address your concerns adequately.

Comments from Reviewer 3

1. *"The authors have addressed my concerns and I am happy to recommend publication. I believe this paper will be impactful, and be of broad interest to researchers who study social relationships and social networks."*

R3.1 Thank you very much for your helpful feedback and suggestions. We are glad that you found the revised paper to be suitable for publication.

2. *"I do have a quibble about R3.2 in the response letter. I don't think one can examine non-linearity by plotting the ranks of the two variables. A monotonic non-linear relationship will appear as linear when transformed into rank orders. While I don't think it will affect the main claims of the manuscript, I do think it will be informative to examine if the relationship is linear or non-linear, and would encourage the authors to consider adding a scatter plot of the variables (either in their original units, or in z-scores) to the supplementary material."*

R3.2 Thank you for this comment. We primarily intended the plots (with rank-transformed in-degree centrality on the horizontal axis and rank-transformed neural similarity on the vertical axis) that we included in point R3.2 of our first response letter to address the question of whether or not there was "something special about being in the top third" of the in-degree centrality distribution. We thought that it would be easiest to see which points corresponded to the top third of the distribution of in-degree centrality in the plot once the horizontal axis had been rank-transformed. We also recognize that the rank-transformed versions of these plots

obfuscate some information about the distribution of in-degree centralities in this sample and about the relationships between in-degree centrality and neural similarities. Therefore, we include plots below that visualize the relationships between centrality and mean neural similarity in three key brain regions (rDMPFC, IDMPFC, and rPCC) with the unranked versions of the variables. We now include both versions of these plots (with and without rank transformation to the horizontal and vertical axes) in the revised supplementary information (as Supplementary Fig. 2, which we also include below).